# PRACTICAL AND RIGOROUS EXTREMAL BOUNDS FOR GAUSSIAN PROCESS REGRESSION VIA CHAINING

## ABSTRACT

Gaussian process regression (GPR) is a popular nonparametric regression method based on Bayesian principles that, unlike most machine learning techniques, provides uncertainty estimates for its predictions. Recent GPR research has focused on enhancing robustness to model misspecification but has often neglected improvements to the underlying methods for computing bounds. In addition, current GPR methods rely heavily on scaling posterior standard deviations and assume well-specified models, both of which reduce GPR's adaptability and accuracy. To address these limitations, we draw inspiration from the chaining method (Talagrand, 2014), and derive chaining bounds for the prediction intervals of GPR, offering a more flexible and accurate approach to handling model uncertainty. Our experimental results validate our theoretical findings, and demonstrate that our method outperforms existing approaches on synthetic and real-world datasets.

## 1 INTRODUCTION

For many applications, especially those requiring safety assurances, obtaining reliable uncertainty estimates is crucial. In this regard, Gaussian process regression (GPR), a flexible non-parametric Bayesian method, is becoming increasingly popular in machine learning fields such as learning-based control methods. GPR assumes that the observed data is generated by a Gaussian process (GP) with independently and identically distributed Gaussian noise. The GP can be fitted to training data and be used to generate predictions along with their associated uncertainty estimates.

Bounds are a way to measure uncertainty in GPR, and both rigorous and practical bounds are the goals of existing research. Wu & Schaback (1993) use the classic method of Fourier transforms to achieve such bounds. By exploiting the properties of reproducing kernel Hilbert spaces (RKHS), Schaback (1999) derives uniform error bounds with faster convergence rates. Relying on an upper bound of the maximum information gain, Srinivas et al. (2009; 2012) and Chowdhury & Gopalan (2017) successively improve methods for frequentist uncertainty bounds.

Given the severe consequences of incorrect hyperparameter specification, recent research has shifted focus to improving robustness. Lederer et al. (2019) introduce probabilistic Lipschitz constants to reduce reliance on prior knowledge. Fiedler et al. (2021) include an error term to modify an objective bound function and improve its resilience to noise. Capone et al. (2022) improve robustness by calculating error bounds based on a given range of hyperparameters. Recently, Papadopoulos (2024) utilizes conformal prediction to calibrate prediction intervals for robustness.

However, this recent line of GPR methods primarily focuses on enhancing robustness but remains constrained by their reliance on scaling the posterior standard deviation, without fundamentally improving their approach to deriving GPR bounds. These approaches also often assume a well-specified model and heavily depend on hyperparameters, which limits their adaptability to new domains and can result in inaccurate error estimates. To address these limitations, we draw inspiration from chain-based techniques (Talagrand, 2014), and propose a chaining-based method. By decomposing the problem into smaller, more refined stages, our method enables more effective error control and improved robustness, especially in complex domains.

In our work, we introduce not only general bounds but also tailored, rigorous bounds for commonly used covariance functions in GPR, such as the Radial Basis Function (RBF) and Matérn kernels. These bounds deepen the theoretical understanding of the kernels' behaviors and enhance their ver-

satility and practicality. Our numerical experiments support our theoretical results and demonstrate the superior performance of our method on both synthetic and real-world datasets.

## 2 BACKGROUND

### 2.1 GAUSSIAN PROCESS REGRESSION

Gaussian process regression (GPR) serves as a robust, non-parametric Bayesian approach to regression (Williams & Rasmussen, 2006). A Gaussian process (GP) defined over an index set or input set $T$ is characterized by a collection of random variables, such that any finite subset has a multivariate normal distribution. In practice, a GP is used to define a distribution over a family of functions $\{f\}$ that could describe the data, and we write $f(x) \sim \mathcal{GP}(m(x), K(x, x'))$ to indicate that the function $f(x)$ is sampled from its corresponding GP. A GP is fully specified by its mean function $m(x)$ (which represents the average value of the functions in the family at each point $x$) and its covariance function $K(x, x')$ (which reflects the extent to which the values of the functions in the family vary together at the points $x$ and $x'$). Popular covariance functions include the radial basis function Kernel (RBF) kernel and Matérn kernel. For simplicity, it is often assumed without loss of generality that $m(x) \equiv 0$.

### 2.2 CHAINING

Chaining is a mathematical technique consisting of a succession of steps that provide successive approximations of an index space $(T, d)$, where $T$ is an index set or input set, and $d$ is a metric on $T$. Its fundamental idea is to group variables $X_t$ that are nearly identical and approximate them at successive levels of granularity (Talagrand, 2014). By doing this, we achieve more effective bounds, especially in cases where many variables are similar (Asadi & Abbe, 2020). This approach mitigates the risk of large errors that can arise from such correlations.

To illustrate, consider a stochastic process $(X_t)_{t \in T}$, and the difference between $X_t$ and $X_{t_0}$ is expressed as $X_t - X_{t_0} = \sum_{n \geq 1} (X_t - X_{t-1})$. When many variables $X_t$ in $T$ are nearly identical, strong correlations between them can obscure the true variation in the process. Grouping similar variables together helps reduce this redundancy by allowing us to approximate these highly correlated variables with a representative value, thereby simplifying the analysis and making the process easier to interpret and work with. A more detailed explanation can be found in Appendix A.

For $n \geq 0$, we select a subset $T_n$, and for each $t \in T$, we choose an approximation $\pi_n(t)$ from $T_n$. Using these $\pi_n(t)$ points, we obtain the corresponding $X_{\pi_n(t)}$ variables, which serve as successive approximations of $X_t$. We start by assuming that $T_0$ contains only one element $t_0$, and thus $\pi_0(t) = t_0$ for all $t \in T$. The core relation is:

$$X_t - X_{t_0} = \sum_{n=1} \left( X_{\pi_n(t)} - X_{\pi_{n-1}(t)} \right).$$

This equality holds because, for sufficiently large $n$, $\pi_n(t)$ equals $t$, meaning that beyond a certain point, the approximation stops, and the series becomes a finite sum. Specifically, as $n$ increases, the sets $T_n$ become progressively finer, eventually covering all points in $T$. Once $T_n$ contains $t$, we have $\pi_n(t) = t$, so no new information is added by further terms in the series. As a result, the infinite series truncates to a finite sum. This ensures convergence in practical settings where the process $X_t$ is fully captured after a finite number of terms.

The efficacy of this approach is rooted in the fact that for each approximation $\pi_i(t)$, the variables $X_t - X_{\pi_i(t)}$ are smaller than $X_t - X_{t_0}$, making their supremum easier to handle. This stepwise refinement converts the intractable global bound estimation into manageable local problems, simplifying the overall calculation. Exponential decay is employed to tighten the bounds through gradual decomposition and layer-by-layer control, thus avoiding the complexity and error accumulation typically associated with global estimation. A more detailed explanation is provided in Appendix A.

## 3 RELATED WORK

The concept of bounds in Gaussian process regression (GPR) originates from the confidence intervals intrinsic to Gaussian processes. This concept is later extended by the bandit literature (Srinivas et al., 2009) to include frequentist uncertainty bounds, which utilize the principle of maximum information gain. Building on this, Srinivas et al. (2012) introduce the use of the reproducing kernel Hilbert space (RKHS) norm for computing bounds in the bandit setting. Thereafter, Chowdhury & Gopalan (2017) significantly advance this line of research, but their work continued to rely on upper bounds given by maximum information gain. To address computational costs, Bartels et al. (2023) propose probabilistic bounds with minimal computational overhead by leveraging intermediate computations performed by the Cholesky decomposition.

Another closely related concept is error bounds, which refer to the absolute gap between the predicted and ground-truth values. In this case, the predicted value plus and minus the error bounds can be regarded as upper and lower bounds respectively. Since the regression produced by radial basis function (RBF) interpolation is equivalent to the GP posterior mean with noiseless training data, classical methods use Fourier transform techniques to derive such error bounds for functions in the reproducing kernel Hilbert space (RKHS) associated with the interpolation kernel (Wu & Schaback, 1993). By further exploiting RKHS properties, uniform error bounds with faster convergence rates are derived by Schaback (1999).

The aforementioned methods assume the accurate specification of the GPR model, using empirical or heuristic approaches to determine its appropriate hyperparameters. However, the *misspecification* of model hyperparameters can have serious consequences. As a result, recent research has focused on improving the robustness of GPR. Lederer et al. (2019) introduce probabilistic Lipschitz constants to reduce prior knowledge, estimating errors on a finite grid and extending them to the entire input space. Fiedler et al. (2021) modify their bound function by introducing an error term based on the work of Chowdhury & Gopalan (2017). Capone et al. (2022) address hyperparameter misspecification by proposing a method to calculate error bounds based on a given hyperparameter range. More recently, Papadopoulos (2024) uses conformal prediction to calibrate prediction intervals using a nonconformity measure to evaluate the degree to which a candidate is unusual or nonconforming.

However, these methods focus solely on robustness to model misspecification and noise, while their underlying approach remains limited to scaling the posterior standard deviation for each instance, without introducing new computational strategies for their bounds. Additionally, existing methods are limited by their reliance on the assumption that the model is well-specified in terms of parameters (e.g., the length scale) and hyperparameters (e.g., noise parameters (Fiedler et al., 2021) and the hyperparameter space (Capone et al., 2022)). This reliance reduces adaptability and often leads to an over- or under-estimation of the bounds. Unlike these methods, our chaining technique mitigates GPR's reliance on the global posterior mean and hyperparameter tuning.

The method proposed by Capone et al. (2022) primarily addresses errors resulting from model misspecification, but it may be less effective when dealing with datasets that have been subjected to added noise. While Fiedler et al. (2021) considers such errors, their approach relies heavily on the selection of the noise-level hyperparameter and is still constrained by the underlying concept of scaling the posterior error. This means that even if the posterior error estimation is accurate, a significant bias in the posterior mean could prevent an adequate coverage of the prediction intervals. (We provide examples and more detailed explanations of these issues in Appendix B.2.) In contrast, our chaining technique ameliorates these issues.

In highly concentrated datasets, such as those involving temporally or spatially continuous data (e.g., temperature time-series), adjacent data points tend to exhibit strong correlations. Traditional methods typically rely on global kernel functions to compute the mean across data points, which makes it difficult to effectively capture such localized correlations. In contrast, chaining methods gather highly correlated data points by defining different layers of approximation, allowing for a layer-by-layer refinement that better controls errors. Additionally, in high-dimensional and complex datasets, where distances between points vary significantly, chaining methods are more adept at capturing local variations, thereby preventing error accumulation and yielding tighter bounds. (We provide examples and more detailed explanations in the Appendix B.3.) In contrast to traditional methods, our technique inherits the aforementioned benefits of chaining to mitigate such problems.

Chain-based methods in machine learning have recently started to gain attention, with Chaining Mutual Information (CMI; (Asadi et al., 2018)) being an example. CMI is a technique that uses mutual information to quantify the shared information between two random variables.This approach has been applied to derive bounds on the expected generalization error of supervised learning algorithms, based on the regularity of the loss function (Clerico et al., 2022). Additionally, CMI has been employed in the context of hierarchical coverings of neural networks to establish risk bounds for neural networks (Asadi & Abbe, 2020). While all these methods apply chaining to the principle of mutual information, in the specific domain of GPR, the covariance function is a more appropriate tool for measuring dependency because it directly defines the structure of the Gaussian process. Therefore, in our work, we apply chaining directly to the covariance functions.

## 4 UPPER AND LOWER BOUNDS

We now present our primary technical contributions. A key observation is that existing methods for uncertainty bounds in Gaussian processes remain largely focused on posterior-based approaches, while chaining techniques have yet to be fully explored in this context. Chaining systematically approximates the upper bound through hierarchical refinements, leveraging incremental estimates between data points. This approach offers greater flexibility and robustness without relying on prior assumptions, especially in high-dimensional spaces. Additionally, chaining excels at capturing local variations in non-smooth processes by refining estimates at each layer. We shall demonstrate that chaining provides rigorous upper bounds through the use of increment and metric entropy techniques, guaranteeing uniform convergence even under noisy conditions and complex metric spaces.

The following theorem, which is a modified version of (Talagrand, 2014), is fundamental for the rest of the paper. In the theorem, we consider a Gaussian process $(X_t)_{t \in T}$ where each $X_t$ is normally distributed with mean zero and variance $\sigma^2$, and $T$ is an index set. ($T$ could also be regarded as an $n$-dimensional input set, e.g., $T \subseteq \mathbb{R}^n$.) For any two points $s, t \in T$, the increment $X_s - X_t$ is given by $E[(X_s - X_t)^2] = d(s,t)^2$, where $d(s,t)$ is a distance metric on $T$. We also make use of the property of a Gaussian distribution that the probability that the absolute increment exceeds a threshold $u$ is bounded by $P(|X_s - X_t| \geq u) \leq 2 \exp\left(-\frac{u^2}{2d(s,t)^2}\right)$.

**Theorem 1.** *(Talagrand, 2014) Let $T$ be an index set, $t_0 \in T$ be an initial index, $T_n \subseteq T$ for $n \geq 0$, and $T_0 = \{t_0\}$. For each $t \in T$, let $\pi_n(t) \in T_n$ for each $n \geq 0$, where each $\pi_n(t)$ represents a successive approximation of $t$, and let $\pi_n(t) = t$ for sufficiently large $n$. Then*

$$P\left(\sup_{t \in T} |X_t - X_{t_0}| > uS\right) \leq L \exp\left(-\frac{u^2}{2}\right), \tag{1}$$

*where $L$ is a universal constant, $u \in \mathbb{R} \cup \{0\}$, $d{:}T{\times}T{\to}\mathbb{R}$ is a distance metric on $T$, and*

$$S := \sup_{t \in T} \sum_{n \geq 1} 2^{n/2} d(\pi_n(t), \pi_{n-1}(t)). \tag{2}$$

It is important to highlight that Theorem 1 is purely theoretical, lacking practical implementation details. For instance, the constant $L$ is introduced without explicit calculation, and no method is provided for determining $S$, $\{T_n\}_{n \geq 0}$, $\{\pi_n(t)\}_{n \geq 0}$, and $t_0$. We address some of these deficiencies below and give a general bound for kernel functions that applies to all kernels.

**Theorem 2.** *(General Bound) Theorem 1, combined with the formula $\mathbb{E}[Y] = \int_0^\infty P(Y \geq u)\, du$, which expresses the expectation, leads to the derivation of the following upper bound for GPR:*

$$\mathbb{E} \sup_{t \in T} X_t \leq X_{t_0} + \mathbb{E}\left[\sup_{t \in T} |X_t - X_{t_0}|\right] \leq X_{t_0} + (1 + \sqrt{2})\sqrt{\frac{\pi}{2}} L \sup_{t \in T} \sum_{n \geq 0} 2^{n/2} d(t, T_n), \tag{3}$$

*where $d(t, T_n)) = \inf_{s \in T_n} \sqrt{K(t,t) + K(s,s) - 2K(t,s)}$ and $t_0$ is chosen such that $X_{t_0}$ is close to zero due to the zero-mean property and the symmetry of the covariance function.*

We provide proofs of Theorem 1 and Theorem 2 in Appendix C.1.

In subsequent sections, we will address these gaps by offering practical implementations with pseudocode. The following subsections apply the general bounds to compute tighter bounds for specific kernels by deriving more precise estimates of $\mathbb{E}[\sup_{t \in T} |X_t - X_{t_0}|]$. We will first introduce the RBF and Matérn kernels, and then provide detailed proofs for their respective tighter bounds.

## 4.1 KERNELS

In Gaussian process regression (GPR), the distance between two input points is typically measured using a kernel function, also commonly known as the covariance function. This function quantifies the similarity between input points in the feature space and plays a pivotal role in defining the Gaussian process structure by influencing the model's smoothness and generalization ability.

One of the most commonly used kernels is the radial basis function (RBF) kernel, also known as the Gaussian kernel. It is favored for its ability to produce smooth and continuous estimates, often in conjunction with a constant kernel to account for signal variance. It is defined as:

$$K(s,t) = \sigma^2 \exp\left(-\frac{\|s-t\|^2}{2l^2}\right),$$

where $\|s-t\|$ is the Euclidean distance between the (multi-dimensional) input points $s$ and $t$, the $\sigma^2$ term represents the constant kernel, and $l$ is the length-scale parameter that controls the smoothness of the function.

The Matérn kernel function, another widely used covariance function in Gaussian processes (GPs), provides a flexible way to model the smoothness of the function being learned. It is defined as:

$$K(s,t) = \frac{2^{1-\nu}}{\Gamma(\nu)} \left(\frac{\sqrt{2\nu}\|s-t\|}{l}\right)^\nu B_\nu\left(\frac{\sqrt{2\nu}\|s-t\|}{l}\right),$$

where $l > 0$ is the length scale parameter, $\|s-t\|$ denotes the Euclidean distance between the input vectors $s$ and $t$, $\Gamma(\cdot)$ is the Gamma function, $B_\nu(\cdot)$ is the modified Bessel function of the second kind, and $\nu > 0$ is a parameter that controls the smoothness of sampled functions.

As $\nu$ increases, the functions sampled from the GP become smoother. The Matérn covariance function becomes simpler when $\nu$ is half-integer: $\nu = p + 1/2$, where $p$ is a non-negative integer (Seeger, 2004). When this happens, the covariance function becomes a product of an exponential and a polynomial of order $p$, with the general expression being: $K(s,t) = \exp\left(-\frac{\sqrt{2\nu}\|s-t\|}{l}\right)\frac{\Gamma(p+1)}{\Gamma(2p+1)}\sum_{i=0}^{p}\frac{(p+i)!}{i!(p-i)!}\left(\frac{\sqrt{8\nu}\|s-t\|}{l}\right)^{p-i}$. In machine learning, one of the most commonly used values for the kernel is $\nu = 3/2$, for which:

$$K(s,t) = \left(1 + \frac{\sqrt{3}\|s-t\|}{l}\right)\exp\left(-\frac{\sqrt{3}\|s-t\|}{l}\right). \tag{4}$$

The distance between two points $s$ and $t$ in the context of GPs is defined as $d(s,t) = \sqrt{\mathbb{E}[(X_s - X_t)^2]}$, where $X_s$ and $X_t$ are the values at points $s$ and $t$ respectively. This distance metric is derived from the covariance function $K(s,t)$, which describes the covariance between the random variables $X_s$ and $X_t$. Specifically, it can be expanded as:

$$d(s,t)^2 = \mathbb{E}[(X_s - X_t)^2] = \mathbb{E}[X_s^2] + \mathbb{E}[X_t^2] - 2\mathbb{E}[X_s X_t] = K(s,s) + K(t,t) - 2K(s,t). \tag{5}$$

It is worth noting that $s$ and $t$ can each represent a vector describing a (multi-dimensional) input in a feature space, with $X_s$ and $X_t$ corresponding to the outputs evaluated at those input vectors. In this case, the covariance function $K(s,t)$ reflects how similar the outputs are given their respective input vectors $s$ and $t$.

## 4.2 TIGHTER BOUNDS FOR RADIAL BASIS FUNCTION (RBF) KERNEL

We will now discuss how to modify the previous bounds in a targeted manner to obtain tighter and more practical upper and lower bounds on Gaussian processes using RBF kernels. This is made precise in the following result. Its detailed proof is provided in Appendix C.4.

**Theorem 3.** *(Tighter RBF Bound) Consider a Gaussian process $(X_t)_{t\in T}$ with a radial basis function (RBF) kernel $K(s,t) = \sigma^2 \exp\left(-\frac{\|s-t\|^2}{2l^2}\right)$, where $T$ is an input/index set, $\|s-t\|$ is the Euclidean distance between input points $s \in T$ and $t \in T$, the term $\sigma^2$ represents the constant kernel, and $l$ represents the length-scale parameter. Let $t_0 \in T$ be an initial point, and $(T_n)_{n\geq 0}$*

*be a sequence such that $T_n \subseteq T$. In addition, for each $t \in T$, let $\{\pi_n(t) \in T_n\}_{n \geq 0}$ represent a chain of successive approximations of $t$ such that $X_t - X_{t_0} = \sum_{n \geq 1} \left( X_{\pi_n(t)} - X_{\pi_{n-1}(t)} \right)$ with the condition that $\pi_n(t) = t$ for sufficiently large $n$ and $\pi_0(t) = t_0$. Then*

$$\mathbb{E} \sup_{t \in T} |X_t - X_{t_0}| \leq (1 + \sqrt{2}) \sqrt{\frac{\pi}{2}} L \sup_{t \in T} \sum_{n \geq 0} 2^{n/2} d'(t, T_n), \tag{6}$$

*where $d'(t, T_n) = \inf_{s \in T_n} \sqrt{K(t,t) + K(s,s) - 2\sigma K^{\frac{1}{2}}(t,s)}$.*

*Proof.* The following inequality holds for $s, t, u \in T$:

$$\|s - t\|^2 + \|t - u\|^2 \geq \frac{(\|s - t\| + \|t - u\|)^2}{2} \geq \frac{\|s - u\|^2}{2}.$$

The first inequality above follows from the Cauchy-Schwarz inequality applied to the special case of two dimensions, while the second inequality follows from the triangle inequality.

Let $x_1 = -\frac{\|s-t\|^2}{l^2}$ and $x_2 = -\frac{\|t-u\|^2}{l^2}$, so that the distance $d(s,u)^2 \leq 2\sigma^2 \left(1 - \exp(a + b)\right)$. Using the Taylor series expansion $\exp(x) = 1 + x + \frac{x^2}{2!} + \frac{x^3}{3!} + \cdots$, we get:

$$\exp(x_1) + \exp(x_2) - 1 = 1 + (x_1 + x_2) + \frac{x_1^2 + x_2^2}{2!} + \frac{x_1^3 + x_2^3}{3!} + \cdots$$

$$\leq 1 + (x_1 + x_2) + \frac{(x_1 + x_2)^2}{2!} + \frac{(x_1 + x_2)^3}{3!} + \cdots = \exp(x_1 + x_2).$$

Using $\exp(x_1) + \exp(x_2) - 1 \leq \exp(x_1 + x_2)$ in the second inequality below, we obtain:

$$d(s,u)^2 \leq 2\sigma^2 \left(1 - \exp(x_1 + x_2)\right) \leq 2\sigma^2 + 2\sigma^2(1 - \exp(x_1) - \exp(x_2))$$

$$= 4\sigma^2 - 2\sigma K^{\frac{1}{2}}(s,t) - 2\sigma K^{\frac{1}{2}}(t,u) = d'(s,t)^2 + d'(t,u)^2,$$

where $d'(s,t)^2 = K(s,s) + K(t,t) - 2\sigma K^{\frac{1}{2}}(s,t)$.

Since $\pi_n(t)$ approximates $t$, it is natural to let:

$$d(t, \pi_n(t)) = d(t, T_n) := \inf_{s \in T_n} d(t, s). \tag{7}$$

With a change of variable $n \to n + 1$, we get:

$$S = \sup_{t \in T} \sum_{n \geq 1} 2^{n/2} d'(\pi_n(t), \pi_{n-1}(t)) \leq (1 + \sqrt{2}) \sup_{t \in T} \sum_{n \geq 0} 2^{n/2} d'(t, T_n).$$

By applying Theorem 1 and Equation 2, the proof is established. A more detailed proof is given in Appendix C.4. □

## 4.3 TIGHTER BOUNDS FOR MATÉRN KERNEL

While the RBF kernel is the most widely used, other kernels, such as the Matérn kernel, are better suited for specific applications. In the following, we provide and prove the upper and lower chaining bounds for the Matérn kernel with its parameter $\nu = 3/2$.

**Theorem 4.** *(Tighter Matérn Bound) Consider a Gaussian process $(X_t)_{t \in T}$ with a Matérn kernel $K(s,t) = \left(1 + \frac{\sqrt{3}\|s-t\|}{l}\right) \exp\left(-\frac{\sqrt{3}\|s-t\|}{l}\right)$ where $T$ is an input/index set, $\|s - t\|$ is the Euclidean distance between input points $s \in T$ and $t \in T$, and $l$ is the length-scale parameter. Let $t_0 \in T$ be an initial point, and $(T_n)_{n \geq 0}$ be a sequence such that $T_n \subseteq T$. In addition, for each $t \in T$, let $\{\pi_n(t) \in T_n\}_{n \geq 0}$ represent a chain of successive approximations of $t$ such that $X_t - X_{t_0} = \sum_{n \geq 1} \left( X_{\pi_n(t)} - X_{\pi_{n-1}(t)} \right)$ with the condition that $\pi_n(t) = t$ for sufficiently large $n$ and $\pi_0(t) = t_0$. Then*

$$\mathbb{E} \sup_{t \in T} |X_t - X_{t_0}| \leq (1 + \sqrt{2}) \sqrt{\frac{\pi}{2}} L \sup_{t \in T} \sum_{n \geq 0} 2^{n/2} [d'(t, T_n) + \sqrt{2} - 2], \tag{8}$$

*where* $\qquad d'(t, T_n)) = \inf_{s \in T_n} \sqrt{K(t,t) + K(s,s) - 2K'(t,s)},$ *and* $K'(s,t) = \left(1 + \frac{\sqrt{3}\|s-t\|}{l}\right) \left[\exp\left(-\frac{\sqrt{3}\|s-t\|}{l}\right) - \frac{1}{2}\right].$

*Proof.* From $K(s,t) = \left(1 + \frac{\sqrt{3}\|s-t\|}{l}\right)\exp\left(-\frac{\sqrt{3}\|s-t\|}{l}\right)$, we get $K(s,s) = K(t,t) = 1$. By substituting $K(s,s) = K(t,t) = 1$ and the kernel function $K(s,t)$ into Equation 5, we obtain: $d(s,t)^2 = 2 - 2\left(1 + \frac{\sqrt{3}\|s-t\|}{l}\right)\exp\left(-\frac{\sqrt{3}\|s-t\|}{l}\right)$.

Using $x_1 = \frac{\sqrt{3}\|s-t\|}{l}$ and $x_2 = \frac{\sqrt{3}\|t-u\|}{l}$, the Chebyshev's sum inequality for $n = 2$ becomes:

$$(1 + x_1)\exp(-x_1) + (1 + x_2)\exp(-x_2) \leq \frac{(1 + x_1 + 1 + x_2)[\exp(-x_1) + \exp(-x_2)]}{2}. \quad (9)$$

Since $\|s-t\| \geq 0$ and $\|t-u\| \geq 0$, we have: $\exp(-x_i) \leq 1$. Using the observation that $(1 - \exp(-x_1))(1 - \exp(-x_2)) > 0$, we get: $\frac{(2+x_1+x_2)}{2}[\exp(-x_1) + \exp(-x_2)] \leq \frac{(2+x_1+x_2)}{2}[1 + \exp(-x_1 - x_2)]$. Negating $\frac{(2+x_1+x_2)}{2}$ and combining with Eq. 9, we get:

$$(1 + x_1)[\exp(-x_1) - \frac{1}{2}] + (1 + x_2)[\exp(-x_2) - \frac{1}{2}] \leq (1 + x_1 + x_2)\exp(-x_1 - x_2).$$

For the function $f(x) = (1 + x)\exp(-x)$, the derivative of $f(x)$ with respect to $x$, calculated using the product rule, is $f'(x) = \frac{d}{dx}[(1 + x)\exp(-x)] = -x\exp(-x)$. Since $\frac{\sqrt{3}\|\mathbf{x}-\mathbf{x}'\|}{l} \geq 0$, we have $f'(x) \leq 0$ (i.e., $f(x)$ is monotonically decreasing) when $x \geq 0$. Using these facts together with the triangle inequality $\|s-t\| + \|t-u\| \geq \|s-u\|$, we get:

$$K(s,u) \geq (1 + x_1)[\exp(-x_1) - \frac{1}{2}] + (1 + x_2)[\exp(-x_2) - \frac{1}{2}] = K'(s,t) + K'(t,u).$$

We can then calculate the distance as:

$$d(s,u)^2 \leq 2 - 2[K'(s,t) + K'(t,u)] = d'(s,t)^2 + d'(t,u)^2 - 2.$$

With a change of variable $n \leftarrow n + 1$), we get:

$$S \leq \sup_{t\in T}\sum_{n\geq 1} 2^{n/2}\sqrt{d'^2(t,T_n) + d'^2(t,T_{n-1}) - 2} \leq (1 + \sqrt{2})\sup_{t\in T}\sum_{n\geq 0} 2^{n/2}(d'(t,T_n) - \frac{\sqrt{2}}{1+\sqrt{2}}).$$

By applying Theorem 1 and Equation 2, the proof is established. A more detailed proof is given in Appendix C.5. $\qquad\square$

By using the bounds from Theorems 3 and 4, we ensure that Gaussian processes have appropriate chaining-based upper bounds. We significantly broaden the applicability of the chaining method, and thus enhance its generalization capacity. The ability to adaptively compute bounds for different kernels, such as the RBF and Matérn kernels, improves the robustness of our approach, making it more versatile in various practical scenarios, especially in high-dimensional and complex domains (as demonstrated by our experimental results in Section 5).

### 4.4 ALGORITHM OF OUR CHAINING METHOD

In this work, we convert theoretical constructs into a practical chaining method for calculating the upper and lower bounds of Gaussian process regression (GPR) with different kernel functions. The full procedure is detailed in Algorithm 1.

First, we preprocess the data by dividing it into training and test sets. Then, we calculate the average of the output values (labels), and center the training set by subtracting the average from the output values of each example (now their mean is 0). Similarly, we subtract the average value from the test set. Next, we fit a Gaussian process (GP) to the training data via maximum likelihood estimation to learn the parameters of the GP's kernel function and ensure that the kernel effectively models the underlying data distribution.

Subsequently, we apply the chaining method to the training set by first constructing the set $T$ containing the features of the training set's examples. As explained in Section 2.2, our objective is to construct a sequence of subsets $T_n$, such that for each $t \in T$, an approximation $\pi_n(t)$ is selected from $T_n$. To obtain more accurate approximations, we iteratively build $T_n$ from the previous subsets $\{T_i\}_{i=1}^{n-1}$, ensuring that $\sup_{t\in T} d(t,T_n)$ is minimized . Specifically, the method iteratively adds the

points farthest from $T_n$ to progressively reduce $\sup_{t \in T} d(t, T_n)$. Thus as the iterations proceed, the approximations become better.

We control the size of the set $T_n$ by using the condition $|T_n| \leq N_n$, where $N_0 = 1$ and $N_n = 2^{2^n}$ for $n \geq 1$. This assumption leverages the approximation $\sqrt{\log N_n} \approx 2^{n/2}$, which is a critical component in our analysis, and is related to the term $\exp(-x^2)$, which governs the tails of a Gaussian distribution. Furthermore, the inequality $N_n^2 \leq N_{n+1}$ demonstrates the effectiveness of this sequence in controlling the size of the sets $T_n$ (Talagrand, 2014).

Next, we compute the distances between the test data and $T_n$, applying Equations 2 and 6 from Theorem 3 or Equation 8 from Theorem 4 to calculate the upper bound $\mathbb{E} \sup_{t \in T} X_t = X_{t_0} + \mathbb{E} \sup_{t \in T} |X_t - X_{t_0}|$. Due to the zero-mean property of the GP and the symmetry of the covariance function, we derive two conclusions: (i) $t_0$ should ideally be chosen such that $X_{t_0}$ is close to zero; otherwise, $\mathbb{E} \sup_{t \in T} |X_t - X_{t_0}|$ would be overestimated; (ii) the infimum can be taken as the negative of the supremum, leading to the lower bound of $\mathbb{E} \inf_{t \in T} X_t = X_{t_0} - \mathbb{E} \sup_{t \in T} |X_t - X_{t_0}|$.

We derive an explicit value for the constant $L$ in Equations 6 and 8 as follows (the derivation is in Appendix C.1):

$$ L = \sum_{n \geq 1} 2 \cdot 2^{2^{n+1}} \exp\left(-2^{n+1}\right) = \sum_{n \geq 1} 2 \left(\frac{2}{e}\right)^{2^{n+1}} . $$

---

**Algorithm 1:** Chaining Bounds Method

---

**Input** : Kernel function $K(s, t)$ and dataset $D := \{(t, X_t)\}$, where $t \in \mathbb{R}^d$ is a $d$-dimensional input/index vector, and $X_t \in \mathbb{R}$ is its associated output value.
**Output:** $B$, a set containing the upper and lower bounds for each test example.
Split $D$ into a training set $D_{\text{train}}$ and a test set $D_{\text{test}}$.
Fit a Gaussian process using the kernel function $K(s, t)$ to the training data $D_{\text{train}}$.
$t_0 \leftarrow \text{argmin}_{t:(t,X_t) \in D_{\text{train}}} |X_t|$
$T_0 \leftarrow \{t_0\}$
$T \leftarrow \{t : (t, \cdot) \in D_{\text{train}}\}$     ($T$ is the set of input/index vectors in $D_{\text{train}}$.)
$n_{\max} \leftarrow \lfloor \log_2(\log_2(|T|)) \rfloor$     ($n_{\max}$ is the largest integer such that $2^{2^{n_{\max}}} < |T|$.)
**for** $n \leftarrow 1$ **to** $n_{max}$ **do**
   $T_n \leftarrow T_{n-1}$
   **while** $|T_n| < 2^{2^n}$ **do**
      $T_n \leftarrow T_n \cup \{\text{argmax}_{t_i \in T} d(t_i, T_n)\}$     ($d(t_i, T_n)$ is computed using Equation 7.)
$B \leftarrow \emptyset$
**foreach** $(t, X_t) \in D_{test}$ **do**
   Compute $\mathbb{E} \sup_t |X_t - X_{t_0}|$ using the set $\{T_n\}_{n=0}^{n_{\max}}$ with Equation 6 from Theorem 3 (RBF kernel) or Equation 8 from Theorem 4 (Matérn kernel).
   $B \leftarrow B \cup \{(X_{t_0} + \mathbb{E} \sup_t |X_t - X_{t_0}|, \ X_{t_0} - \mathbb{E} \sup_t |X_t - X_{t_0}|)\}$
**return** $B$

---

## 5 EXPERIMENT

### 5.1 DATASETS

- **Synthetic Data.** This dataset is generated by producing 50 random functions from a Reproducing Kernel Hilbert Space (RKHS) over the domain $D = [-1, 1]$, evaluated at 1000 evenly spaced points. Each function is constructed by combining kernel functions centered at randomly selected points. For each function, we sample 50 input values and add Gaussian noise with a standard deviation of 0.5.

- **Boston House Price** (Cournapeau et al., 2007). This dataset contains the median house prices for 506 areas in Boston, MA, USA. Each area is described by 13 input features (e.g., crime rates and pollution), with the median house price for that area as the target variable.

- **NOAA Weather** (NOAA, 2020). This dataset provides daily weather summaries from various locations, featuring multiple variables such as wind speed, humidity, and precipitation. The objective is to predict temperature.
- **Sarcos** (Schaal, 2009). This dataset contains recordings from a seven-degree-of-freedom robotic arm, with 21 input features representing joint positions, velocities, and accelerations. The goal is to predict the required torque for each of the seven joints.

## 5.2 EVALUATION METRICS

The performance of our proposed approach is evaluated using standard metrics for prediction intervals, as described by (Khosravi et al., 2010).

- **Prediction Interval Coverage Probability (PICP)**. This metric evaluates the percentage of test observations that lie within the bounds of the prediction intervals (PIs) . It is calculated as $PICP = \frac{1}{n} \sum_{i=1}^{n} c_i$, where $c_i = 1$ if the output at point $i$ lies within the bounds $[L(X_i), U(X_i)]$, and $c_i = 0$ otherwise. Here, $L(X_i)$ and $U(X_i)$ denote the lower and upper bounds of the $i^{th}$ PI.
- **Normalized Mean Prediction Interval Width (NMPIW)**. PIs that are too wide provide little useful information, so the NMPIW metric quantifies the width of the PIs as:

$$NMPIW = \frac{\frac{1}{n} \sum_{i=1}^{n} (U(X_i) - L(X_i))}{R},$$

where $R$ is the range of the target variable. NMPIW expresses the average PI width as a percentage of the target range.

- **Coverage Width-Based Criterion (CWC)**. This is the *primary* evaluation metric because it balances the conflicting goals of achieving narrow PIs (low NMPIW) and high coverage (high PICP). (Note that a good PICP score can be trivially achieved at the expense of NMPIW (by using overly wide PIs) and vice versa (by using overly narrow PIs). Hence either PICP or NMPIW alone is insufficient to completely reflect the goodness of bounds.) CWC is defined as:

$$CWC = NMPIW \left(1 + \gamma(PICP)e^{-\eta(PICP - \mu)}\right),$$

where $\gamma$ and $\eta$ are hyperparameters, and $\mu$ represents the nominal confidence level ($\mu = 1$ for extremal bounds). When $PICP \geq \mu$, $\gamma = 0$; otherwise, $\gamma = 1$.

## 5.3 BASELINES

We compare our chaining method to the following three state-of-the-art baselines that are described in Section 3: (i) **Lederer19** (Lederer et al., 2019), which introduces probabilistic Lipschitz constants to reduce the reliance on prior knowledge, estimates errors on a finite grid, and extends them to the input space; (ii) **Fiedler21** (Fiedler et al., 2021), which modifies its objective bound function by introducing an error term based on the work of (Chowdhury & Gopalan, 2017); and (iii) **Capone22** (Capone et al., 2022), which tackles hyperparameter misspecification by proposing a method to calculate error bounds across a given range of hyperparameters.

## 5.4 RESULTS

Table 1 compares the performances of our method and the baselines. Achieving high PICP is important for ensuring that the predicted intervals capture the true outcomes. In terms of this metric, both our method and Fiedler21 consistently deliver strong performance. Our method achieves perfect coverage across all datasets, while Fiedler21 attains near-perfect results (it slightly underperforms under the higher noise condition of the synthetic data).

While narrower PIs are desirable for improving NMPIW, excessively tight intervals can compromise coverage and thus PICP. Lederer19 and Capone22 frequently produce narrower intervals (NMPIW) but often at the cost of inadequate coverage (PICP).

Note that CWC is the primary metric of evaluation because it combines and effectively balances the competing demands of the other two. In terms of CWC, our method consistently performs the

best by achieving the lowest CWC. This indicates that our method has superior coverage while maintaining compact intervals. The baselines often struggle to balance coverage and interval width, particularly on the synthetic dataset where noise results in under-coverage and lower PICP.

| | Synthetic Data | | | Boston House Prices | | |
|---|---|---|---|---|---|---|
| | PICP(↑) | NMPIW(↓) | **CWC(↓)** | PICP(↑) | NMPIW(↓) | **CWC(↓)** |
| RBF(Ours) | **1.00** | 2.53 | **2.53** | **1.00** | 2.12 | **2.12** |
| Matérn(Ours) | **1.00** | 3.67 | 3.67 | **1.00** | 2.78 | 2.78 |
| Capone22 | 0.54 | **0.58** | 5.69e+09 | 0.49 | **0.09** | 7.92e+09 |
| Fiedler21 | 0.99 | 1.53 | 3.48 | **1.00** | 3.46 | 3.46 |
| Lederer19 | 0.94 | 0.78 | 16.48 | 0.80 | 0.55 | 7.67e+03 |
| | Sarcos | | | NOAA Weather | | |
| | PICP(↑) | NMPIW(↓) | **CWC(↓)** | PICP(↑) | NMPIW(↓) | **CWC(↓)** |
| RBF(Ours) | **1.00** | 0.75 | **0.75** | **1.00** | 3.67 | **3.67** |
| Matérn(Ours) | **1.00** | 1.14 | 1.14 | **1.00** | 6.52 | 6.52 |
| Capone22 | 0.60 | 0.04 | 1.40e+07 | **1.00** | 8.80 | 8.80 |
| Fiedler21 | **1.00** | 1.42 | 1.42 | **1.00** | 9.31 | 9.31 |
| Lederer19 | 0.93 | **0.12** | 4.10 | 0.94 | **0.21** | 5.23 |

Table 1: Comparison of Our Method against Baselines on Synthetic and Real-world Datasets.

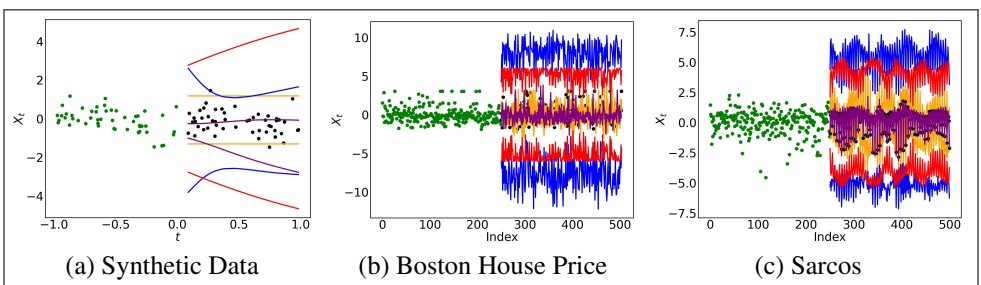

(a) Synthetic Data          (b) Boston House Price          (c) Sarcos

Figure 1: Comparison of Our Method with Baselines. The training set is in green, the test set in black, Lederer19 in orange, Fiedler21 in blue, Capone22 in purple, and our method in red.

Figure 1 illustrates Table 1. In all plots, our method achieves 100% coverage (all black test points are within our bounds) with narrower bounds on average, demonstrating its superior performance over all baselines. The next best system, Fiedler21, have upper and lower bounds (blue lines) that perform moderately well overall but occasionally under- or over-estimate compared to our method. For example, in Figure 1(a), one test point remains uncovered. In Figure 1(b) and Figure 1(c), observe that Lederer19 and Capone22 do not cover all the black test points, while Fiedler21 and our method do. However, our method does so with tighter bounds than Fiedler21. (Bigger and clearer plots, and more empirical results are provided in Appendix B.)

## 6 CONCLUSION

Our work addresses the limitations of existing Gaussian Process Regression methods by introducing a novel chain-based approach that improves error control and robustness. By leveraging Talagrand's techniques (Talagrand, 2014) and developing rigorous bounds for commonly used kernels, such as RBF and Matérn, we advance both the theoretical foundations and practical application of these models. The superior performance of our method, empirically demonstrated across both synthetic and real-world datasets, underscores its effectiveness in enhancing prediction accuracy and uncertainty quantification in GPR. As future work, we would like to extend our approach to more kernels and to optimizing regret in multi-arm bandit problems.

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

## A    REVIEW OF CHAINING

Next we review at a high level the scheme of the chaining bound method.

The goal is to bound $\mathbb{E}Y$ where $Y = \sup_t(X_t - X_{t_0})$. We introduce a "ood set" $\Omega_u$ for a given parameter $u \geq 0$, which excludes undesirable events. As $u$ becomes large, $P(\Omega_u^c)$ becomes small. When $\Omega_u$ occurs, we bound $Y$, say $Y \leq f(u)$, where $f$ is an increasing function on $\mathbb{R}_+$.

$$\mathbb{E}Y = \int_0^\infty P(Y \geq u)du \leq f(0) + \int_0^\infty P(Y \geq f(u))du,$$

$$\mathbb{E}Y = f(0) + \int_0^\infty f'(u)P(Y \geq f(u))du,$$

where we have used a change of variables in the last equality. Now, since $Y \leq f(u)$ on $\Omega_u$, we have:

$$P(Y \geq f(u)) \leq P(\Omega_u^c),$$

and finally:

$$\mathbb{E}Y \leq f(0) + \int_0^\infty f'(u)P(\Omega_u^c)du.$$

In practice, we will always have $P(\Omega_u^c) \leq L \exp(-u/L)$ and $f(u) = A + u^\alpha B$, yielding the bound:

$$\mathbb{E}Y \leq A + K(\alpha)B.$$

At the heart of this example is the introduction of a "good set" $\Omega_u$, which confines undesirable events to a small probability. As the parameter $u$ increases, the probability of bad events $\Omega_u^c$ decreases exponentially. This allows for effective error control within the "good set," avoiding the coarse global error estimates typically used in traditional methods.

Furthermore, by controlling tail probabilities and utilizing exponential decay bounds, such as $P(\Omega_u^c) \leq L \exp(-u/L)$, along with the function $f(u) = A + u^\alpha B$, the chaining method ensures that the final error remains well-controlled. This level of probabilistic precision, achieved by breaking the problem into layers and managing each incremental error independently, prevents the overestimation of total error that is common in traditional approaches.

## B    MORE EXPERIMENTAL RESULTS AND ANALYSIS

### B.1    COMPUTATIONAL COST AND SCALABILITY

The proposed method has three primary computational steps: fitting the Gaussian process, constructing the sets $\{T_n\}$, and computing bounds for the test points. Fitting the Gaussian process involves matrix factorization with a complexity of $\mathcal{O}(|D_{\text{train}}|^3)$. Constructing $\{T_n\}$ requires $\mathcal{O}(|D_{\text{train}}|^2 \cdot \log\log|D_{\text{train}}|)$, dominated by kernel distance computations. Finally, computing bounds for $|D_{\text{test}}|$ test points has a complexity of $\mathcal{O}(|D_{\text{test}}| \cdot |D_{\text{train}}| \cdot \log\log|D_{\text{train}}|)$. The total computational complexity depends on the relative sizes of the training and test sets. Since the sizes of the training and test sets can vary, the overall complexity is determined by the more computationally intensive step. Thus, the total time complexity is: $\mathcal{O}(\max(|D_{\text{train}}|^2 \cdot \log\log|D_{\text{train}}|, |D_{\text{test}}| \cdot |D_{\text{train}}| \cdot \log\log|D_{\text{train}}|))$.

We also evaluated computational cost and scalability, with the table detailing data size and runtime for each method. All numerical experiments in this section were conducted on a Linux system with kernel version 5.15.0-112-generic (#122-Ubuntu SMP Thu May 23 07:48:21 UTC 2024). The machine configuration includes an x86_64 processor with 16 CPU cores and 125.49 GB of RAM.

| | Synthetic Data | Boston House Price | NOAA Weather | Sarcos |
|---|---|---|---|---|
| Train Data Size | 50 | 250 | 255 | 250 |
| Test Data Size | 50 | 254 | 110 | 4000 |

Table 2: Size of Datasets.

For computational cost, our methods (RBF and Matérn) perform competitively across various datasets. On smaller datasets like Synthetic Data and NOAA Weather, RBF and Matérn achieve

| Time(s) | Synthetic Data | Boston House Price | NOAA Weather | Sarcos |
|---|---|---|---|---|
| RBF(Ours) | 0.05 | **0.52** | 1.95 | 1.74 |
| Matérn(Ours) | **0.04** | 0.77 | **0.40** | 1.75 |
| Capone22 | 30.68 | 149.63 | 5.83 | 343.54 |
| Fiedler21 | 0.07 | 0.75 | 1.34 | **1.18** |
| Lederer19 | 0.56 | 2.31 | 1.92 | 2.86 |

Table 3: Computational Cost and Scalability of Our Method with Baselinesin Synthetic Data.

notably lower runtimes than other methods; RBF, for instance, requires only 0.0524 seconds on Synthetic Data and 1.9504 seconds on NOAA Weather, while Matérn achieves the lowest runtime on NOAA Weather at 0.4038 seconds. However, on larger datasets such as Boston House Price and Sarcos, Fiedler21 demonstrates a computational advantage, with a runtime of 1.1845 seconds on Sarcos, outperforming both RBF (1.7396 seconds) and Matérn (1.7545 seconds). Thus, while methods perform optimally at different dataset scales, RBF and Matérn are particularly effective for small to medium datasets, while Fiedler21 shows greater efficiency with large datasets.

Scalability was assessed by examining performance across increasingly large datasets. RBF and Matérn exhibit robust scalability, maintaining controlled runtime growth even with substantial dataset increases, especially on the Sarcos dataset. This stable performance underscores their adaptability to larger datasets with minimal efficiency loss. Fiedler21 also scales well, with competitive runtime on large datasets (1.1845 seconds on Sarcos), making it suitable for large-scale applications. In contrast, Capone22's runtime increases sharply with data size, indicating limited scalability and reduced practicality for very large datasets. Lederer19 demonstrates moderate scalability, performing well on medium to large datasets but showing some limitations as data size expands.

## B.2 SYNTHETIC DATA

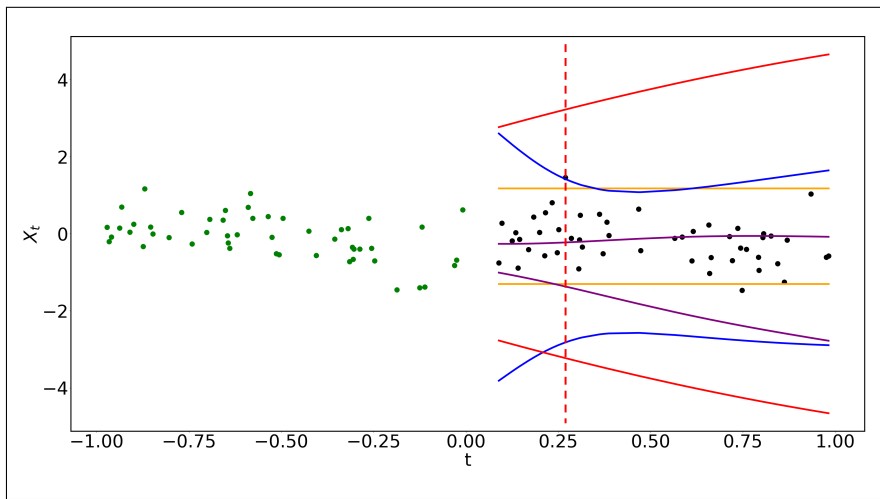

Figure 2: Comparison of Our Method with Baselinesin Synthetic Data. The training set is in green, the test set in black, Lederer19 in orange, Fiedler21 in blue, Capone22 in purple, and our method in red.

We compare the advantages of our method to prior approaches using an example from the experimental dataset in Figure 2, which includes significant noise at a level of 0.5. We focus on a key point with a true value of $1.458$, which all other methods failed to capture within their prediction intervals.

Capone22 predicts $-0.79 \pm 0.57$, underestimating the true value, likely due to an inadequate treatment of the dataset's noise. This can be attributed to Capone22's focus on model misspecification

errors rather than noise impact on the prediction bounds, leading to poor performance in this instance.

The Lederer19 method provides bounds of [-1.30, 1.18], which are insufficient, and while Fiedler21 performs better, it still does not fully encompass the true value. The Fiedler21 method predicts $-0.69 \pm 2.12$, with the posterior variance increased due to the noise level being correctly set at 0.5—something we know because this is a generated dataset. This hyperparameter significantly impacts error magnitude, and while we use the correct noise level here, incorrect tuning would lead to even worse predictions in other cases. However, the posterior mean is still too low, causing a slight underestimation of the true value. Although the model accounts for significant uncertainty, its reliance on the posterior mean skews the prediction bounds.

In contrast, our chain method does not rely on fixed noise parameters. Instead, it progressively refines the posterior estimate by breaking the process down into layers, with each layer capturing different local variations in the data. This multi-scale approach reduces the noise's impact on the posterior mean by spreading the uncertainty across multiple levels. As a result, the chain method produces a prediction interval that is not only more accurate but also successfully encompasses the true value. The broader uncertainty range reflects a more realistic variance estimation, avoiding the overly tight bounds seen in other methods, which tend to shrink the variance too much and underestimate the true uncertainty.

## B.3 REAL-WORLD DATA

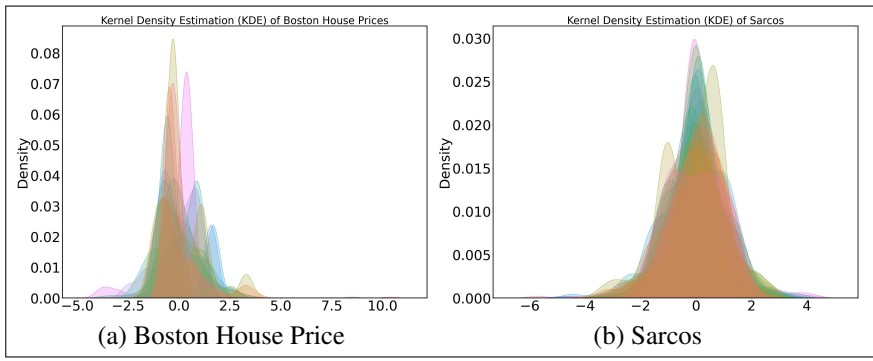

(a) Boston House Price  (b) Sarcos

Figure 3: KDE of Boston House Price Data and Sarcos Data.

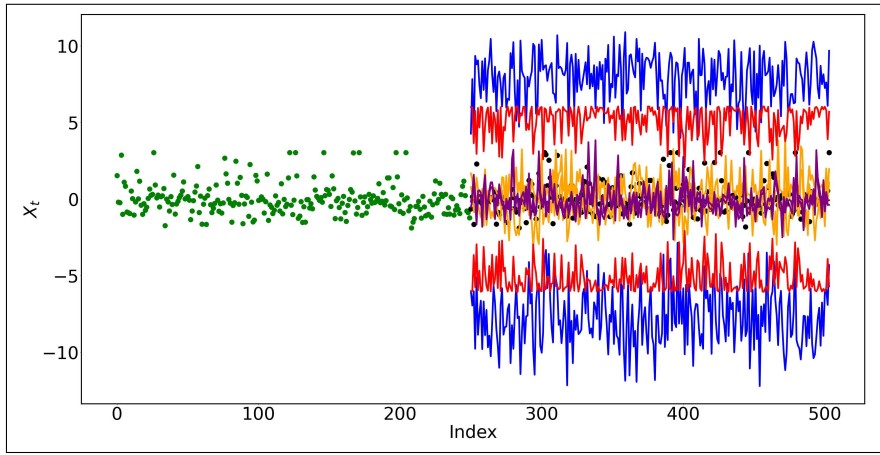

Figure 4: Comparison of Our Method with Baselines in Boston House Price Data. The training set is in green, the test set in black, Lederer19 in orange, Fiedler21 in blue, Capone22 in purple, and our method in red.

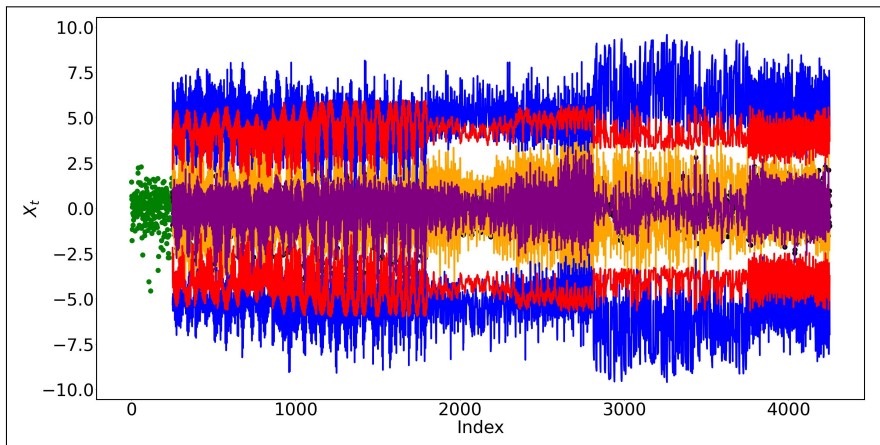

Figure 5: Comparison of Our Method with Baselines in Sarcos Data. The training set is in green, the test set in black, Lederer19 in orange, Fiedler21 in blue, Capone22 in purple, and our method in red.

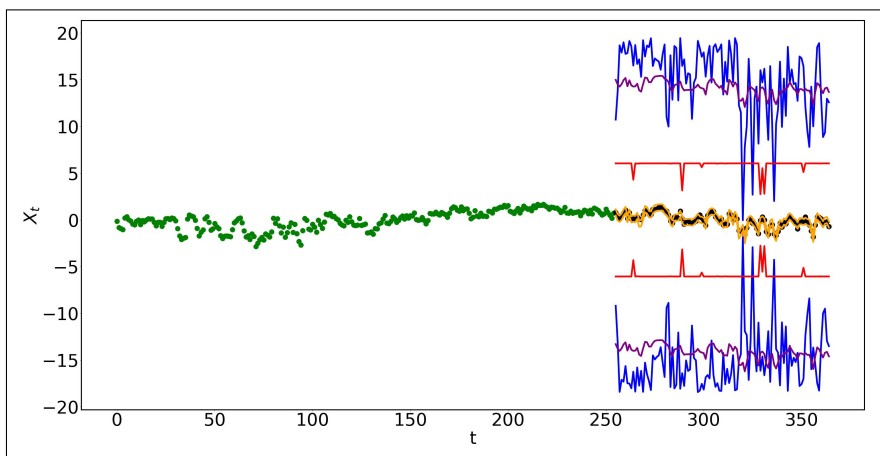

Figure 6: Comparison of Our Method with Baselines in NOAA Weather Data. The training set is in green, and test set in black, Lederer19 in orange, Fiedler21 in blue, Capone22 in purple, and our method in red. Observe that Lederer19's bounds do not cover all the test data points whereas our method and other baselines do. Also note that compared to the other baselines that cover all test points (Fiedler21 and Capone 22), our method has the tightest bounds.

We also consider two real-world datasets, viz. Boston House Price and Sarcos, both of which exhibit highly concentrated, high-dimensional, and complex characteristics. As shown in Figure 3, the kernel density estimation (KDE) plots for these datasets display sharp peaks, indicating the highly correlated nature of the data and their complexity across multiple dimensions.

Traditional methods typically rely on the entire kernel function to compute the mean of the data points, which makes it difficult to handle such strong local correlations effectively. In contrast, the chaining method groups data in highly correlated regions together by defining successive approximation layers, refining the approximation step by step, and thus controlling the error. Additionally, in high-dimensional, complex data, where distances between points can vary significantly and in more complex ways, the chaining method captures local variations more effectively, preventing the accumulation of errors and resulting in tighter bounds.

Figures 4 and Figure 5 illustrate this point. In these datasets, the bounds obtained by Fiedler21 are notably wider compared to those produced by our chaining method (Fiedler21 is the next best technique compared to our method in Table 1). This demonstrates how the chaining method excels

in controlling error and achieving more precise bounds in the context of highly concentrated data, where traditional methods like Fiedler21 struggle to maintain accuracy.

## B.4 STATISTICAL SIGNIFICANCE

To evaluate the statistical significance of our method compared to the baseline models (Fiedler21, Capone22, and Lederer19), we performed paired t-tests on the CWC metric. For each dataset (Boston House Price, NOAA Weather, and Sarcos), the training and testing sets were randomly sampled 100 times. In each trial, the models were trained on the training set and evaluated on the testing set, resulting in 100 independent CWC values for each model.

The paired t-tests were then applied to these CWC values to compare our method with the baselines. This approach ensures that the comparisons account for the variability introduced by the random splits while maintaining the dependency between paired observations. Since lower CWC values indicate better performance, negative t-statistics demonstrate that our method consistently outperformed the baselines. We used $p < 0.01$ to denote high statistical significance and $p < 0.05$ for moderate significance.

| Model Comparison | t-Statistic | p-Value | Statistical Significance |
|---|---|---|---|
| Our Method vs Fiedler21 | -16.39 | <0.001 | ** |
| Our Method vs Capone22 | -45.48 | <0.001 | ** |
| Our Method vs Lederer19 | -10.61 | <0.001 | ** |

Table 4: Paired t-Test Comparisons of Our Method against Baselines on the Boston House Price Data. (** indicates $p < 0.01$; * indicates $p < 0.05$; negative t-statistics indicate that our model performs better than the compared model, as lower CWC values are preferable.)

| Model Comparison | t-Statistic | p-Value | Statistical Significance |
|---|---|---|---|
| Our Method vs Fiedler21 | -89.87 | <0.001 | ** |
| Our Method vs Capone22 | -63.54 | <0.001 | ** |
| Our Method vs Lederer19 | -32.39 | <0.001 | ** |

Table 5: Paired t-Test Comparisons of Our Method against Baselines on the NOAA Weather Data. (** indicates $p < 0.01$;* indicates $p < 0.05$; negative t-statistics indicate that our model performs better than the compared model, as lower CWC values are preferable.)

| Model Comparison | t-Statistic | p-Value | Statistical Significance |
|---|---|---|---|
| Our Method vs Fiedler21 | -3.64 | <0.001 | ** |
| Our Method vs Capone22 | -177.71 | <0.001 | ** |
| Our Method vs Lederer19 | -15.88 | <0.001 | ** |

Table 6: Paired t-Test Comparisons of Our Method against Baselines on the Sarcos Data. (** indicates $p < 0.01$; * indicates $p < 0.05$; negative t-statistics indicate that our model performs better than the compared model, as lower CWC values are preferable.)

The paired t-test results, detailed in Tables 4, 5, and 6, demonstrate significant differences between our method and the baselines. For the Boston House Price dataset, our method outperformed all baselines with high statistical significance ($p < 0.01$), supported by negative t-statistics, as lower CWC values indicate better performance. On the NOAA Weather dataset, significant differences were consistently observed ($p < 0.01$ for all comparisons). Similarly, for the Sarcos dataset, our method showed statistically significant improvements ($p < 0.01$) across all baselines. These results strongly emphasize the statistical significance of our method's performance advantages.

## B.5 INTERPOLATION (AKA INFILL)

Theoretically, our method relies solely on the kernel to compute distances, making it applicable to both extrapolation and interpolation tasks. This is because the kernel function quantifies the similarity between data points based on their relative positions, independent of whether the points

lie within or outside the observed range. As a result, the method naturally generalizes to scenarios where test points are interpolated within the training set.

To empirically validate this, we conducted an interpolation experiment using NOAA data. The horizontal axis represents time, with the middle 70% of the data used as the test set (black points) and the leftmost and rightmost 30% as the training set (green points). The red lines represent the computed bounds. As shown in the figure, the bounds successfully encompass all test points, achieving a PICP of 1.0. The NMPIW is 1.6545 and the CWC is also 1.6545, highlighting that our method is well-suited for interpolation tasks, providing tight and reliable bounds while maintaining theoretical consistency with its kernel-based design.

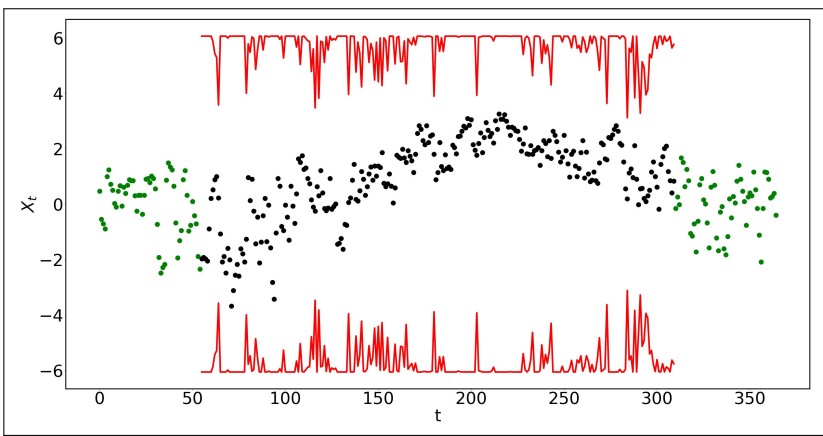

Figure 7: Interpolation experiment on NOAA Weather Data. The training set is shown in green, the test set in black, and the bounds predicted by our method in red.

## C  PROOF OF THEOREMS

### C.1  PROOF OF THEOREM 1

This theorem and its proof are due to (Talagrand, 2014). We provide a modified, more compact version to aid in exposition and intuition building. For the complete proof, please refer to (Talagrand, 2014).

Assume that $(X_t)_{t \in T}$ is a Gaussian process, where each $X_t$ is normally distributed with mean zero. For any two points $s, t \in T$, the increment $X_s - X_t$ is given by:

$$E[(X_s - X_t)^2] = d(s, t)^2,$$

where $d(s, t)$ is a distance metric on $T$.

Given a normally distributed random variable $Z$ with mean zero and variance $\sigma^2$, the probability that $|Z|$ exceeds a threshold $u$ is bounded by: $P(|Z| \geq u) \leq 2 \exp\left(-\frac{u^2}{2\sigma^2}\right)$. Applying this result to the increment $X_s - X_t$, we substitute $\sigma^2$ with $d(s, t)^2$ and get:

$$P(|X_s - X_t| \geq u) \leq 2 \exp\left(-\frac{u^2}{2d(s,t)^2}\right).$$

This implies the expresion below when $u = u2^{n/2}d(\pi_n(t), \pi_{n-1}(t)))$:

$$P(|X_{\pi_n(t)} - X_{\pi_{n-1}(t)}| \geq u2^{n/2}d(\pi_n(t), \pi_{n-1}(t))) \leq 2 \exp\left(-u^2 2^{n-1}\right)$$

The number of possible pairs $(\pi_n(t), \pi_{n-1}(t))$ is bounded by:

$$|T_n| \cdot |T_{n-1}| \leq N_n N_{n-1} \leq N_{n+1} = 2^{2^{n+1}}.$$

We define the (favorable) event $\Omega_{u,n}$ by

$$\forall t, \ |X_{\pi_n(t)} - X_{\pi_{n-1}(t)}| \leq u2^{n/2}d(\pi_n(t), \pi_{n-1}(t)),$$

and we define $\Omega_u = \bigcap_{n \geq 1} \Omega_{u,n}$. Then

$$p(u) := P(\Omega_u^c) \leq \sum_{n \geq 1} P(\Omega_{u,n}^c) \leq \sum_{n \geq 1} 2 \cdot 2^{2^{n+1}} \exp(-u^2 2^{n-1}).$$

Here again, at the crucial step, we have used the union bound $P(\Omega_u^c) \leq \sum_{n \geq 1} P(\Omega_{u,n}^c)$. When $\Omega_u$ occurs, it yields

$$|X_t - X_{t_0}| \leq u \sum_{n \geq 1} 2^{n/2}d(\pi_n(t), \pi_{n-1}(t)),$$

so that

$$\sup_{t \in T} |X_t - X_{t_0}| \leq uS,$$

where

$$S := \sup_{t \in T} \sum_{n \geq 1} 2^{n/2}d(\pi_n(t), \pi_{n-1}(t)).$$

Thus,

$$P\left(\sup_{t \in T} |X_t - X_{t_0}| > uS\right) \leq p(u).$$

Given $n \geq 1$ and $u \geq 3$, the series can be bounded by

$$u^2 2^{n-1} \geq \frac{u^2}{2} + u^2 2^{n-2} \geq \frac{u^2}{2} + 2^{n+1}.$$

For

$$p(u) \leq L \exp\left(-\frac{u^2}{2}\right),$$

we observe that since $p(u) \leq 1$, the inequality holds not only for $u \geq 3$ but also for $u > 0$, because $1 \leq \exp(\frac{9}{2}) \exp\left(-\frac{u^2}{2} - 2^{n+1}\right)$ for $u \leq 3$. Hence,

$$P\left(\sup_{t \in T} |X_t - X_{t_0}| \geq uS\right) \leq L \exp\left(-\frac{u^2}{2}\right)$$

where $L$ is an constant term. $\square$

## C.2 DERIVATION OF THE VALUE FOR $L$

From the proof of Theorem 1, we have:

$$p(u) \leq \sum_{n \geq 1} 2 \cdot 2^{2^{n+1}} \exp\left(-\frac{u^2}{2} - 2^{n+1}\right).$$

Thus,

$$L = \sum_{n \geq 1} 2 \cdot 2^{2^{n+1}} \exp\left(-2^{n+1}\right) = \sum_{n \geq 1} 2 \left(\frac{2}{e}\right)^{2^{n+1}}. \qquad \square$$

## C.3 PROOF OF THEOREM 2

Given any $t_0$ in $T$, the centering hypothesis implies

$$E \sup_{t \in T} X_t = E \sup_{t \in T}(X_t - X_{t_0}).$$

The latter form has the advantage that we now seek estimates for the expectation of the nonnegative random variable $Y = \sup_{t \in T}(X_t - X_{t_0})$. For such a variable, we have the formula

$$EY = \int_0^\infty P(Y \geq u) \, du.$$

Using Theorem 1:

$$P\left(\sup_{t \in T} |X_t - X_{t_0}| \geq uS\right) \leq L \exp\left(-\frac{u^2}{2}\right)$$

From it, to perform the integration, we introduce a new variable $v$. Let $v = \frac{u}{S}$, then $du = S dv$. Thus,

$$\mathbb{E}\left[\sup_{t \in T} |X_t - X_{t_0}|\right] \leq L \cdot \int_0^\infty \exp\left(-\frac{v^2}{2}\right) S dv.$$

Simplifying, we get:

$$\mathbb{E}\left[\sup_{t \in T} |X_t - X_{t_0}|\right] \leq LS \int_0^\infty \exp\left(-\frac{v^2}{2}\right) dv,$$

where

$$S := \sup_{t \in T} \sum_{n \geq 1} 2^{n/2} d(\pi_n(t), \pi_{n-1}(t)).$$

This integral is a standard Gaussian integral, and the result is:

$$\int_0^\infty \exp\left(-\frac{v^2}{2}\right) dv = \sqrt{\frac{\pi}{2}}.$$

Since $\pi_n(t)$ approximates $t$, it is natural to assume that:

$$d(t, \pi_n(t)) = d(t, T_n) := \inf_{s \in T_n} d(t, s).$$

The triangle inequality yields:

$$d(\pi_n(t), \pi_{n-1}(t)) \leq d(t, \pi_n(t)) + d(t, \pi_{n-1}(t)) = d(t, T_n) + d(t, T_{n-1}),$$

so that (making the change of variable Making the change of variable $n \leftarrow n+1$ in the second sum below, we obtain:

$$S = \sup_{t \in T} \sum_{n \geq 1} 2^{n/2} d(\pi_n(t), \pi_{n-1}(t))$$

$$\leq \sup_{t \in T} \sum_{n \geq 1} 2^{n/2} d(t, T_n) + \sup_{t \in T} \sum_{n \geq 1} 2^{n/2} d(t, T_{n-1})$$

$$= \sup_{t \in T} \sum_{n \geq 0} 2^{n/2} d'(t, T_n) + \sqrt{2} \sup_{t \in T} \sum_{n \geq 1} 2^{(n-1)/2} d(t, T_{n-1})$$

$$= \sup_{t \in T} \sum_{n \geq 0} 2^{n/2} d'(t, T_n) + \sqrt{2} \sup_{t \in T} \sum_{n \geq 0} 2^{n/2} d(t, T_n)$$

$$\leq (1 + \sqrt{2}) \sup_{t \in T} \sum_{n \geq 0} 2^{n/2} d(t, T_n).$$

Thus, the result is:

$$\mathbb{E}\left[\sup_{t \in T} |X_t - X_{t_0}|\right] \leq (1 + \sqrt{2})\sqrt{\frac{\pi}{2}} L \sup_{t \in T} \sum_{n \geq 0} 2^{n/2} d(t, T_n).$$

Since

$$\mathbb{E} \sup_{t \in T} X_t \leq \mathbb{E}[X_{t_0}] + \mathbb{E}\left[\sup_{t \in T} |X_t - X_{t_0}|\right] = X_{t_0} + \mathbb{E}\left[\sup_{t \in T} |X_t - X_{t_0}|\right]$$

so that

$$\mathbb{E} \sup_{t \in T} X_t \leq X_{t_0} + \mathbb{E}\left[\sup_{t \in T} |X_t - X_{t_0}|\right] \leq X_{t_0} + (1 + \sqrt{2})\sqrt{\frac{\pi}{2}} L \sup_{t \in T} \sum_{n \geq 0} 2^{n/2} d(t, T_n). \tag{10}$$

where $d(t, T_n)) = \inf_{s \in T_n} \sqrt{K(t,t) + K(s,s) - 2K(t,s)}$ $\quad \square$ .

### C.4 PROOF OF THEOREM 3

A common kernel used in GPR is the radial basis function (RBF) kernel, also known as the Gaussian kernel. In this context, we consider a composite kernel that combines a constant kernel with an RBF kernel. The constant kernel $\sigma^2$ adds a constant variance to the covariance matrix, helping to control the overall amplitude of the process. The combined kernel function is expressed as:

$$K(s, t) = \sigma^2 \exp\left(-\frac{\|s - t\|^2}{2l^2}\right).$$

By substituting $K(s,s) = K(t,t) = 1$ and the kernel function $K(s,t)$ into the distance formula, we obtain:

$$d(s,t)^2 = 2\sigma^2\left(1 - \exp\left(-\frac{\|s - t\|^2}{2l^2}\right)\right).$$

Using the Cauchy-Schwarz inequality In two-dimensional space, we get:

$$\frac{\|s - t\|^2 + \|t - u\|^2}{2} \geq \left(\frac{\|s - t\| + \|t - u\|}{2}\right)^2.$$

Combined with the triangle inequality $\|s - t\| + \|t - u\| \geq \|s - u\|$, we then obtain:

$$\|s - t\|^2 + \|t - u\|^2 \geq \frac{\|s - u\|^2}{2}.$$

Thus the distance is:

$$d(s, u)^2 \leq 2\sigma^2\left(1 - \exp\left(-\frac{\|s - t\|^2 + \|t - u\|^2}{l^2}\right)\right).$$

Recall that the Taylor series expansion of $\exp(x)$ is:

$$\exp(x) = 1 + x + \frac{x^2}{2!} + \frac{x^3}{3!} + \cdots .$$

Let $x_1 = -\frac{\|s-t\|^2}{l^2}$ and $x_2 = -\frac{\|t-u\|^2}{l^2}$. We then get:

$$\exp(x_1) + \exp(x_2) - 1 = 1 + (x_1 + x_2) + \frac{x_1^2 + x_2^2}{2!} + \frac{x_1^3 + x_2^3}{3!} + \cdots$$

$$\leq 1 + (x_1 + x_2) + \frac{(x_1 + x_2)^2}{2!} + \frac{(x_1 + x_2)^3}{3!} + \cdots = \exp(x_1 + x_2).$$

For this inequality, we provide another simpler proof: Given that $x_1, x_2 \geq 0$, it follows that $\exp(x_1) \geq 1$ and $\exp(x_2) \geq 1$. Therefore, $(1 - \exp(x_1))(1 - \exp(x_2)) \geq 0$, i.e., $1 - \exp(x_1) - \exp(x_2) + \exp(x_1 + x_2) \geq 0$.

By using this, we have:

$$d(s, u)^2 = 2\sigma^2 \left(1 - \exp\left(x_1 + x_2\right)\right)$$

$$\leq 2\sigma^2 + 2\sigma^2(1 - \exp\left(x_1\right) - \exp\left(x_2\right))$$

$$= 2\sigma^2(2 - \exp^{\frac{1}{2}}\left(-\frac{\|s-t\|^2}{2l^2}\right) - \exp^{\frac{1}{2}}\left(-\frac{\|t-u\|^2}{l^2}\right))$$

$$= 4\sigma^2 - 2\sigma K^{\frac{1}{2}}(s, t) - 2\sigma K^{\frac{1}{2}}(t, u)$$

$$= 2\sigma^2 - 2\sigma K^{\frac{1}{2}}(s, t) + 2\sigma^2 - 2\sigma K^{\frac{1}{2}}(t, u)$$

$$= d'(s, t)^2 + d'(t, u)^2.$$

where $d'(s, t)^2 = K(s, s) + K(t, t) - 2\sigma K^{\frac{1}{2}}(s, t)$.

Since $\pi_n(t)$ approximates $t$, it is natural to assume that:

$$d(t, \pi_n(t)) = d(t, T_n) := \inf_{s \in T_n} d(t, s).$$

For an RBF kernel, we have:

$$d(s, u)^2 \leq d'^2(s, t) + d'^2(t, u),$$

where $d'(s, t)^2 = K(s, s) + K(t, t) - 2\sigma K^{\frac{1}{2}}(s, t)$.

Making the change of variable $n \leftarrow n + 1$ in the second sum below, we obtain:

$$S = \sup_{t \in T} \sum_{n \geq 1} 2^{n/2} d(\pi_n(t), \pi_{n-1}(t))$$

$$\leq \sup_{t \in T} \sum_{n \geq 1} 2^{n/2} d'(t, T_n) + \sup_{t \in T} \sum_{n \geq 1} 2^{n/2} d'(t, T_{n-1})$$

$$= \sup_{t \in T} \sum_{n \geq 0} 2^{n/2} d'(t, T_n) + \sqrt{2} \sup_{t \in T} \sum_{n \geq 1} 2^{(n-1)/2} d'(t, T_{n-1})$$

$$= \sup_{t \in T} \sum_{n \geq 0} 2^{n/2} d'(t, T_n) + \sqrt{2} \sup_{t \in T} \sum_{n \geq 0} 2^{n/2} d'(t, T_n)$$

$$\leq (1 + \sqrt{2}) \sup_{t \in T} \sum_{n \geq 0} 2^{n/2} d'(t, T_n).$$

Using Equation 2, we obtain the fundamental bound:

$$\mathbb{E} \sup_{t \in T} |X_t - X_{t_0}| \leq (1 + \sqrt{2}) \sqrt{\frac{\pi}{2}} L \sup_{t \in T} \sum_{n \geq 0} 2^{n/2} d'(t, T_n),$$

where

$$d'(t, T_n)) = \inf_{s \in T_n} \sqrt{K(t, t) + K(s, s) - 2\sigma K^{\frac{1}{2}}(t, s)}. \qquad \square$$

## C.5 Proof of Theorem 4

Since $K(s,t) = \left(1 + \frac{\sqrt{3}\|s-t\|}{l}\right)\exp\left(-\frac{\sqrt{3}\|s-t\|}{l}\right)$, we have $K(s,s) = K(t,t) = 1$.

By substituting $K(s,s) = K(t,t) = 1$ and the kernel function $K(s,t)$ into the distance formula, we obtain:

$$d(s,t)^2 = 2 - 2\left(1 + \frac{\sqrt{3}\|s-t\|}{l}\right)\exp\left(-\frac{\sqrt{3}\|s-t\|}{l}\right).$$

The Chebyshev's sum inequality is a fundamental result in the theory of inequalities. It states that if $a_1, a_2$ and $b_1, b_2$ are two sequences of real numbers that are sorted in opposite orders (one in increasing and the other in decreasing order), then the following inequality holds:

$$\frac{1}{n}\sum_{i=1}^{n} a_i b_i \leq \left(\frac{1}{n}\sum_{i=1}^{n} a_i\right)\left(\frac{1}{n}\sum_{i=1}^{n} b_i\right).$$

Specifically, for $a_i = 1 + x_i$ and $b_i = \exp(-x_i)$, which are oppositely sorted, let $x_1 = \frac{\sqrt{3}\|s-t\|}{l}$ and $x_2 = \frac{\sqrt{3}\|t-u\|}{l}$. Then the inequality for $n = 2$ becomes:

$$(1 + x_1)\exp(-x_1) + (1 + x_2)\exp(-x_2) \leq \frac{(1 + x_1 + 1 + x_2)[\exp(-x_1) + \exp(-x_2)]}{2}.$$

Since $\|s - t\| \geq 0$ and $\|t - u\| \geq 0$, we have $\exp(-x_i) \leq 1$. Oberve that $(1 - \exp(-x_1))(1 - \exp(-x_2)) > 0$. Rearranging terms, we obtain:

$$\exp(-x_1) + \exp(-x_2) < 1 + \exp(-x_1)\exp(-x_2) = 1 + \exp(-x_1 - x_2).$$

Using this, we get:

$$(1 + x_1)\exp(-x_1) + (1 + x_2)\exp(-x_2) \leq \frac{(2 + x_1 + x_2)}{2}[\exp(-x_1) + \exp(-x_2)]$$
$$\leq \frac{(2 + x_1 + x_2)}{2}[1 + \exp(-x_1 - x_2)].$$

After negating $\frac{(2+x_1+x_2)}{2}$, we get:

$$(1 + x_1)[\exp(-x_1) - \frac{1}{2}] + (1 + x_2)[\exp(-x_2) - \frac{1}{2}] \leq (1 + x_1 + x_2)\exp(-x_1 - x_2).$$

Given the function $f(x) = (1 + x)\exp(-x)$, the derivative of $f(x)$ with respect to $x$ is calculated using the product rule as:

$$f'(x) = \frac{d}{dx}\left[(1 + x)\exp(-x)\right] = -x\exp(-x).$$

Since $\frac{\sqrt{3}\|s-t\|}{l} \geq 0$, we know that $f'(x) \leq 0$ when $x \geq 0$. Thus $f(x)$ is monotonically decreasing when $n \geq 0$.

With the triangle inequality $\|s-t\| + \|t-u\| \geq \|s-u\|$, and since $f(x)$ is monotonically decreasing, we get:

$$K(s,u) = \left(1 + \frac{\sqrt{3}\|s-u\|}{l}\right)\exp\left(-\frac{\sqrt{3}\|s-u\|}{l}\right)$$
$$\geq (1 + x_1 + x_2)\exp(-x_1 - x_2)$$
$$\geq (1 + x_1)[\exp(-x_1) - \frac{1}{2}] + (1 + x_2)[\exp(-x_2) - \frac{1}{2}]$$
$$= K'(s,t) + K'(t,u),$$

where $K'(s,t) = \left(1 + \frac{\sqrt{3}\|s-t\|}{l}\right)\left[\exp\left(-\frac{\sqrt{3}\|s-t\|}{l}\right) - \frac{1}{2}\right]$.

We can then calculate the distance:

$$
\begin{aligned}
d(s,u)^2 &= K(s,s) + K(u,u) - 2K(s,u) \\
&\leq 2 - 2[K'(s,t) + K'(t,u)] = 2 - 2K'(s,t) + 2 - 2K'(t,u) - 2 \\
&= d'(s,t)^2 + d'(t,u)^2 - 2.
\end{aligned}
$$

For the Matérn kernel (with $v = \frac{3}{2}$), we have proven that:

$$
d(s,u)^2 \leq d'(s,t)^2 + d'(t,u)^2 - 2,
$$

where $d'(s,t)^2 = K(s,s) + K(t,t) - 2K'(s,t)$.

Making the change of variable $n \leftarrow n + 1$ in the second sum below, we get:

$$
\begin{aligned}
S &\leq \sup_{t \in T} \sum_{n \geq 1} 2^{n/2}\sqrt{d'^2(t,T_n) + d'^2(t,T_{n-1}) - 2} \\
&\leq \sup_{t \in T} \sum_{n \geq 1} 2^{n/2}d'(t,T_n) + \sqrt{2}\sup_{t \in T}\sum_{n \geq 0} 2^{n/2}d'(t,T_n) - \sum_{n \geq 0} 2^{n/2}\sqrt{2} \\
&\leq (1+\sqrt{2})\sup_{t \in T}\sum_{n \geq 0} 2^{n/2}\left(d'(t,T_n) - \frac{\sqrt{2}}{1+\sqrt{2}}\right).
\end{aligned}
$$

Using Equation 2, we have the bound:

$$
\mathbb{E}\sup_{t \in T}|X_t - X_{t_0}| \leq (1+\sqrt{2})\sqrt{\frac{\pi}{2}}L\sup_{t \in T}\sum_{n \geq 0} 2^{n/2}[d'(t,T_n) + \sqrt{2} - 2],
$$

where $\qquad d'(t,T_n)) = \inf_{s \in T_n}\sqrt{K(t,t) + K(s,s) - 2K'(t,s)}, \qquad$ and $K'(s,t) = \left(1 + \frac{\sqrt{3}\|s-t\|}{l}\right)\left[\exp\left(-\frac{\sqrt{3}\|s-t\|}{l}\right) - \frac{1}{2}\right]. \qquad \square$

