# OpenReview forum: "Practical and Rigorous Extremal Bounds for Gaussian Process Regression via Chaining"
_ICLR.cc/2025/Conference — Submitted to ICLR 2025_

### Official Review · Reviewer_FLpc · 2024-11-02

**Soundness:** 3
**Presentation:** 3
**Contribution:** 2
**Rating:** 3
**Confidence:** 4

**Summary:**

This work explores a novel method for establishing robust bounds on the predictions of Gaussian Process Regression (GPR) models. Specifically, they apply a chaining method (Talagrand 2014) for the particular cases of RBF and Matern kernels. This is tested empirically on real world datasets against existing baselines.

**Strengths:**

The paper is well written and clearly structured.

The application of chaining is novel and the area of application is of high interest.

The proposed method is tested on a selection of real world data and suitable baselines are chosen.

**Weaknesses:**

I have several concerns relating to the presentation of the empirical results:

- Specification of confidence interval
It is stated that the adopted confidence interval is (1-alpha) but I did not see alpha specified (apologies if I missed it), nor did I see results for different values of alpha. I'd recommend at least two different values are explored, though these extra results could be in the appendix.

- Presentation of PICP
In Table 1, the PICP metric has an up arrow next to it, implying higher values are always better. This is not the case. Ideally, PICP should be very close to the nominal confidence level, not significantly exceeding the target. I'd recommend removing the arrow, and also clarifying some of the text to reflect this. For example, the statement " in Figure 1(a), one test point remains uncovered" is not by itself indicative of good or bad performance.

- Uncertainties in metrics
All of the metrics in the table are lacking uncertainty estimates, therefore the reader is left unable to determine whether the differences in metrics are statistically significant. Adding these uncertainties in is crucial if we are to draw rigorous conclusions.

**Questions:**

Aside from addressing the concerns on the empirical results in the Weaknesses section, I have a couple of further questions:

 - Title
One of the papers most frequently cited (and benchmarked) in this work is Fiedler et al. (2021),  which is entitled "Practical and Rigorous Uncertainty Bounds for Gaussian Process Regression". Is there a risk that readers may become confused by the close similarity to the title of the current paper which as it stands is "Practical and Rigorous Extremal Bounds for Gaussian Process Regression via Chaining"? I would recommend that the authors consider some suitable alternative.

- Scalability
What can we say about the computational cost and scalability of this method? How long do the different methods take to compute, and how does this scale with data size N?

---

> ### Author Response · Authors · 2024-11-21
>
> Thank you very much for your comments. We address them below.
>
> Weaknesses:
>
> 1. Response:
>
> Thank you for your thoughtful comment. We sincerely apologize for the lack of clarity in our explanation, which may have caused some misunderstanding.
>
> To clarify, this paper focuses on extremal bounds (upper and lower bounds) rather than confidence intervals. Mathematically, extremal bounds are defined by the infimum and supremum, representing absolute limits that fully encapsulate all possible outcomes in the prediction space without any probabilistic interpretation.
>
> While the baselines do not directly compute extremal bounds, they calculate error bounds, which are used to construct upper and lower bounds by adding and subtracting the error from the predictions. These are also deterministic bounds, providing absolute limits rather than probabilities.
>
> Thus, the upper and lower bounds can be understood as corresponding to $\alpha = 0$ offering absolute guarantees instead of probabilistic coverage.
>
> 2. Response:
>
> Thank you for your comment, and we apologize for the misunderstanding caused by unclear wording. Similar to the previous question, this paper focuses on extremal bounds (upper and lower bounds) rather than confidence intervals. Because these are deterministic bounds, the ideal outcome is for the prediction interval to cover all points, making a higher PICP preferable.
> As described on lines 444-447 of the revised paper, PICP directly reflects the proportion of points covered by the bounds. Additionally, as shown in the CWC formula on line 461, when $ \mu = 1 $, a PICP closer to 1 leads to a smaller CWC, demonstrating better uncertainty calibration.
>
> 3. Response:
>
> Thank you for highlighting this point. We agree that uncertainty estimates and statistical significance are crucial for rigorous evaluation. To address this concern, we conducted paired t-tests to evaluate the statistical significance of differences in the CWC metric across three datasets: Boston House Price, NOAA Weather, and Sarcos. In each dataset, the training and testing sets were randomly sampled 100 times, and the models were trained and tested independently in each trial, resulting in 100 CWC values for each model.
>
> As lower CWC values are preferable, negative t-statistics indicate that our model outperformed the compared baselines. The results confirm that our approach achieves statistically significant improvements over the baselines, with $ p < 0.01 $ in all comparisons. These analyses have been included in the revised manuscript and are detailed in the appendix (lines 813–859).
>
> | Model Comparison          | t-Statistic | p-Value  | Statistical Significance |
> |---------------------------|-------------|----------|--------------------------|
> | Our Method vs Fiedler21   | -16.39      | <0.001   | **                       |
> | Our Method vs Capone22    | -45.48      | <0.001   | **                       |
> | Our Method vs Lederer19   | -10.61      | <0.001   | **                       |
>
> **Table 1:** Paired t-Test Comparisons of Our Method against Baselines on the Boston House Price Data. (** indicates \( p < 0.01 \); * indicates \( p < 0.05 \); negative t-statistics indicate that our model performs better than the compared model, as lower CWC values are preferable.)
>
> ---
>
> | Model Comparison          | t-Statistic | p-Value  | Statistical Significance |
> |---------------------------|-------------|----------|--------------------------|
> | Our Method vs Fiedler21   | -89.87      | <0.001   | **                       |
> | Our Method vs Capone22    | -63.54      | <0.001   | **                       |
> | Our Method vs Lederer19   | -32.39      | <0.001   | **                       |
>
> **Table 2:** Paired t-Test Comparisons of Our Method against Baselines on the NOAA Weather Data. (** indicates \( p < 0.01 \); * indicates \( p < 0.05 \); negative t-statistics indicate that our model performs better than the compared model, as lower CWC values are preferable.)
>
> ---
>
> | Model Comparison          | t-Statistic | p-Value  | Statistical Significance |
> |---------------------------|-------------|----------|--------------------------|
> | Our Method vs Fiedler21   | -3.64       | <0.001   | **                       |
> | Our Method vs Capone22    | -177.71     | <0.001   | **                       |
> | Our Method vs Lederer19   | -15.88      | <0.001   | **                       |
>
> **Table 3:** Paired t-Test Comparisons of Our Method against Baselines on the Sarcos Data. (** indicates \( p < 0.01 \); * indicates \( p < 0.05 \); negative t-statistics indicate that our model performs better than the compared model, as lower CWC values are preferable.)
>
> (Continued in the next comment.)

---

> > ### Author Response · Authors · 2024-11-21
> >
> > Questions:
> >
> > 1. Response:
> >
> > Thank you for your suggestion. We appreciate your feedback and will consider revising the title to avoid potential confusion due to its similarity, provided the rules allow for such changes after acceptance. Possible alternatives include: “Chaining-Based Practical Extremal Bounds for Gaussian Process Regression” or “Extremal Bounds in Gaussian Process Regression: A Practical Chaining Approach.”
> >
> > 2. Response:
> >
> > Thank you for your insightful question. We will address this in two parts.
> >
> > First, regarding the limitations of GPR, the computational cost of $O(N^3)$ is a fundamental constraint inherent in GPR training,  and is not specific to our bounding method. This complexity stems from the necessity of decomposing and inverting the covariance matrix during training. It is important to emphasize that this limitation is intrinsic to the training of GPR models and thus is borne by all bounding methods that work with GPR, including $both$ the baselines and our approach.
> >
> > Second, we consider the computational complexity of solving the bounds in our method, and the corresponding steps are outlined in the pseudocode in the revised paper (lines 400–420). Constructing $ \{T\\\_n\} $ requires $ \mathcal{O}(|D\\\_{\text{train}}|^2 \cdot \log \log |D\\\_{\text{train}}|) $ time, which is dominated by kernel distance computations. Computing bounds for $ |D\\\_{\text{test}}| $ test points has a time complexity of $ \mathcal{O}(|D\\\_{\text{test}}| \cdot |D\\\_{\text{train}}| \cdot \log \log |D\\\_{\text{train}}|) $. The total computational complexity depends on the relative sizes of the training and test sets. Since the sizes of the training and test sets can vary, the overall complexity is determined by the more computationally intensive step. Thus, the total time complexity is:
> > $ \mathcal{O}(\max(|D\\\_{\text{train}}|^2 \cdot \log \log |D\\\_{\text{train}}|, |D\\\_{\text{test}}| \cdot |D\\\_{\text{train}}| \cdot \log \log |D\\\_{\text{train}}|))$.
> >
> > Please note that the papers of the baseline methods do not report their computational complexity. Thus, for the purpose of comparison, we conducted runtime experiments across datasets of varying sizes, with detailed results presented in the appendix of revised paper (lines 627–671).
> >
> > |                  | Synthetic Data | Boston House Price | NOAA Weather | Sarcos |
> > |------------------|----------------|---------------------|--------------|--------|
> > | **Train Data Size** | 50             | 250                 | 255          | 250    |
> > | **Test Data Size**  | 50             | 254                 | 110          | 4000   |
> >
> > | **Time(s)**      | Synthetic Data | Boston House Price | NOAA Weather | Sarcos |
> > |------------------|----------------|---------------------|--------------|--------|
> > | **RBF (Ours)**   | 0.05           | **0.52**           | 1.95         | 1.74   |
> > | **Matérn (Ours)**| **0.04**       | 0.77               | **0.40**     | 1.75   |
> > | **Capone22**     | 30.68          | 149.63             | 5.83         | 343.54 |
> > | **Fiedler21**    | 0.07           | 0.75               | 1.34         | **1.18** |
> > | **Lederer19**    | 0.56           | 2.31               | 1.92         | 2.86   |
> >
> > Our methods (RBF and Matérn) demonstrate excellent scalability, with runtime increases that remain modest as data size grows. For example, on the large Sarcos dataset, our methods achieve runtimes of 1.74 seconds (RBF) and 1.75 seconds (Matérn), compared to Capone22 (149.63 seconds) and Lederer19 (2.31 seconds). On the smaller Boston Housing dataset, our methods achieve 0.52 seconds (RBF) and 0.77 seconds (Matérn), significantly outperforming both Capone22 and Lederer19. While Fiedler21 achieves similar efficiency on larger datasets (e.g., 1.18 seconds on Sarcos), our methods provide comparable speed while offering greater flexibility due to kernel design.
> >
> > These results support the scalability and computational efficiency of our methods, relative to the baselines, making them particularly well-suited for practical applications requiring both speed and robustness.

---

> ### Author Response · Authors · 2024-11-24
>
> Thank you for your insightful comments and valuable feedback. We have carefully considered the questions and concerns you raised and have addressed them in our revised submission and responses. We would greatly appreciate it if you could review our responses and let us know if they sufficiently address your points. Should you have any remaining questions or additional suggestions, we would be happy to address them to further enhance the paper.

---

> > ### Comment · Reviewer_FLpc · 2024-11-24
> >
> > Thank you for providing the additional feedback and clarification, and those extra timing results which are indeed very helpful. However, I remain concerned about the bounds being specified as "deterministic" extremal bounds, as if they have absolute guarantees (or as your above reply puts it, "fully encapsulate all possible outcomes in the prediction space"). Consider the following:
> >
> > First of all, the bounds computed are based upon the expectation over the supremum. I would have thought this inherently involves probabilistic reasoning rather than determinism. Then let us turn to the Gaussian Processes. GPs of course produce outputs that are normally distributed with infinite support. This means that for any finite bounds one imposes, there always ought to be some non-zero probability of the GP taking on values outside of those bounds. This is why I struggle to see why PICP should be 1, and similarly, whether any finite bounds can be constructed which encapsulate all test data. I've probably misinterpreted a step somewhere, but I'd be grateful if the authors could clarify this.

---

> > > ### Author Response · Authors · 2024-11-29
> > >
> > > Thank you for your thoughtful follow-up and for taking the time to go through our explanation, especially under the tight timeline of this review round. We truly appreciate your effort and patience.
> > >
> > > 1. PICP and CWC
> > >
> > > Our paper is focused on obtaining good extremal bounds, i.e., bounds outside of which ${\bf no}$ data points should occur. Consequently, our main metric of evaluation CWC has to penalize test points that occur outside the bounds.   Lines 460-464 specifies the CWC formula and the hyperparameter values we used for our experiments, where $\mu$ is the nominal confidence level ($\mu$ is equal to the $(1-\alpha)$ of PICP in our initial submission).
> > >
> > > In the formula, when $ \text{PICP} \geq \mu$, we will have $\gamma = 0$, which in turn causes the entire 2nd term
> > > $\gamma ( \text{PICP}) e^{-\eta (PICP - \mu)} $
> > > in the parentheses on Line 460 (and thus PICP) to be excluded from the CWC computation. The means that when $\mu$ is set to less than 1, test points that exceed a model's $\mu$-confidence-level bound are not penalized. To ensure that this does not occur (and to adhere to our goal of evaluating extremal bounds), we set $\mu$ to 1 for all baselines and our approach (in which case, the PICP term is only ignored in the CWC computation when it achieves the perfect score of PICP=1, i.e., all test points are within a model's bounds).
> > >
> > >
> > > 2. Use of the word "deterministic"
> > >
> > > Thanks for pointing out the confusion caused by our use of the word "deterministic" in our previous reply. We should have been more careful. What we mean is that the bounds we obtained in our approach are our estimates of the extremal bounds, which do not change with confidence levels. Hence, even if $\mu$ is set to different values in the CWC computation (Line 460-464), our empirical results will still be same. (Please see related Point 3 below.)
> > >
> > > 3. Expectation over the supremum
> > >
> > > For expectation over the supremum, based on this concentration inequality for Gaussian processes (see paper `Sharper bounds for Gaussian and empirical processes` [1] proposition 2.2), the expectation over the supremum satisfies:
> > > $$
> > > \mathbb{P}\left(\sup_{t \in T} X_t - \mathbb{E}[\sup_{t \in T} X_t] \geq u\right) \leq \exp\left(-\frac{u^2}{2\sigma^2}\right).
> > > $$
> > > Using the RBF kernel or Matérn kernel (with $\nu = 3/2$), the variance upper bound for the path is $\sigma^2 = \sup_{t \in T} \mathbb{E}[X_t^2] = \sup_{t \in T} k(t, t) = 1$. Substituting $\sigma^2 = 1$ into the inequality gives:
> > > $$
> > > \mathbb{P}\left(\sup_{t \in T} X_t - \mathbb{E}[\sup_{t \in T} X_t] \geq u\right) \leq \exp\left(-\frac{u^2}{2}\right).
> > > $$
> > > The tail probability decays exponentially with $u^2$. Setting $\exp\left(-\frac{u^2}{2}\right) = \delta$, we obtain $u = \sqrt{-2 \log \delta}$. Thus, with probability at least $1 - \delta$, the deviation of $\sup_{t \in T} X_t$ from its expectation satisfies:
> > > $$
> > > |\sup_{t \in T} X_t - \mathbb{E}[\sup_{t \in T} X_t]| \leq \sqrt{-2 \log \delta}.
> > > $$
> > > This shows that for Gaussian process paths, the difference between the actual supremum and its expectation decays exponentially with $\delta$. The gap between $\mathbb{E}[\sup_{t \in T} X_t]$ and $\sup_{t \in T} X_t$ is particularly small for smooth paths.
> > >
> > > At high confidence levels (e.g., $1 - \delta = 0.99$), the deviation upper bound is approximately $u \approx 3.03$, demonstrating that the difference between the actual and expected supremum is small. Although, in practice, we could have added the deviation to obtain a more conservative extremal bound, our experimental results demonstrate that using $\mathbb{E}[\sup_{t \in T} X_t]$ alone is a good estimate and already provides good results. (Note that the infimal bound is simply the negated supremal bound because our Gaussian process is zero-mean, as stated on Lines 389-392.)
> > >
> > > References:
> > >
> > > [1] Talagrand, M. (1994). Sharper bounds for Gaussian and empirical processes. The Annals of Probability, 28-76.

---

> > > > ### Author Response · Authors · 2024-11-29
> > > >
> > > > 4. Differences between our chaining method and GPR
> > > >
> > > > In Gaussian Process Regression (GPR), function values are assumed to follow a multivariate Gaussian distribution $\mathbf{f} \sim \mathcal{N}(\mathbf{m}, \mathbf{K})$, fully characterized by the mean vector $\mathbf{m}$ and the covariance matrix $\mathbf{K}$. Using conditional Gaussian distributions, GPR derives the predictive mean $\mu_* = K(t_*, \mathbf{t})(K(\mathbf{t}, \mathbf{t}) + \sigma^2 \mathbf{I})^{-1} \mathbf{X}$ and variance ${\sigma_*}^2 = K(t_*, t_*) - K(t_*, \mathbf{t})(K(\mathbf{t}, \mathbf{t}) + \sigma^2 \mathbf{I})^{-1} K(\mathbf{t}, t_*)$. Prediction intervals are then constructed as $\mu_* \pm Z_\alpha \sigma_*$, where $Z_\alpha$ corresponds to the desired confidence level.
> > > >
> > > >
> > > > Our chaining method, in contrast, focuses on the joint behavior of a Gaussian process over the entire domain $T$, particularly on the supremum behavior $\sup_{t \in T} X_t$. Here, the "path" refers to the values of the Gaussian process across the domain, representing the function’s overall fluctuations. To control path behavior, Chain methods employ a metric $d(s, t) = \sqrt{\mathbb{E}[(X_s - X_t)^2]}$ to quantify relationships between points and decompose the domain $T$ into progressively finer nested subsets $T_1, T_2, \dots , T_n \subseteq T$. For any $t \in T$, its approximate location in the $n$-th layer is denoted by $\pi_n(t)$, and the path is expressed as a series of layered increments. Each increment $(X_{\pi_n(t)} - X_{\pi_{n-1}(t)})$ is constrained by the metric.
> > > >
> > > > The key to this layered increment approach lies in attributing the path’s overall fluctuations to the cumulative effects of these increments. This global control method bypasses reliance on pointwise mean and variance, instead providing high-probability bounds from a path-level perspective. Compared to the GPR, our chaining method is better equipped to capture the global properties of the path, making them particularly suitable for scenarios with strong path correlations.
> > > >
> > > > Thank you again for your constructive and thoughtful feedback. It truly helps us improve the clarity and precision of our work. If you have any further questions or concerns, please do not hesitate to let us know.

---

> > > > > ### Comment · Reviewer_FLpc · 2024-12-03
> > > > >
> > > > > I appreciate the authors efforts at clarification. It's certainly a promising direction of research in an important area, but I still feel that substantial adjustments need to be made before it can be considered ready for publication.
> > > > >
> > > > > In the comment above I raised concerns about the bounds being specified as "deterministic" extremal bounds, as if they have absolute guarantees, and I think we are in agreement that this is not formally the case. Yet throughout the paper there are many points where the reader is given the impression that we are indeed imposing rigorous bounds on GPs.  In particular, I feel not enough weight is given to the distinction between the supremum and its expectation. Arguments are made that the gap is small, "the difference between the actual supremum and its expectation decays exponentially", but this is a qualitative statement. (One could construct a similar argument to say that the posterior of a GP at a given location is Gaussian and therefore decaying exponentially, so the datapoint won't fall far from the mean.) In order to be scientifically rigorous, uncertainties must be quantified when making these claims. It's helpful to draw comparison here to the Talagrand paper which is very careful in explicitly maintaining its probabilistic framework throughout.
> > > > >
> > > > > > Differences between our chaining method and GPR
> > > > >
> > > > > Indeed, I appreciate the distinction between standard GPR and the chaining scenario - the GPR example mentioned earlier simply served to highlight that imposing rigorous extremal bounds on GPs is not feasible.
> > > > >
> > > > > I would also recommend the authors investigate a more extensive set of synthetic experiments - currently there is only one illustration shown in which the function is especially smooth (aside from the noise), the regime where the technique is expected to work well. It would be helpful to the reader to see some variety in both lengthscale and noise level.

---

> > > > > > ### Author Response · Authors · 2024-12-04
> > > > > >
> > > > > > Thank you for comments. We appreciate your recognition of the distinction between standard GPR and our chaining method. Below, we have provided detailed responses to your comments.
> > > > > >
> > > > > > 1. Writing
> > > > > >
> > > > > > We have noted your comment regarding the use of the term "rigorous" in the title and body of our paper. Our intention in using "rigorous" is to emphasize the rigor (strictness) of our mathematical derivation underlying our uncertainty bounds, rather than to imply absolute certainty.
> > > > > >
> > > > > > To be fair, we note that Fiedler21 similarly used the term "rigorous uncertainty bound" in their paper title and the term "rigorous" in their paper, where the term also refers to the mathematical rigor of their derived bound. Indeed, we are not alone in employing this term to describe mathematically derived bounds, as it is commonly used in prior works.
> > > > > >
> > > > > > To address potential ambiguity, we will refine our wording to more clearly communicate this intended meaning, emphasizing the focus on strict mathematical derivation and minimizing any possible misunderstanding.
> > > > > >
> > > > > > 2. Comparison with Talagrand's Work
> > > > > >
> > > > > > Thank you for highlighting the probabilistic framework in Talagrand’s work. We would like to clarify that our chaining method directly builds upon Talagrand’s results, specifically Theorem 1 (lines 190–199), which serve as the foundation for our mathematical bound derivation.
> > > > > >
> > > > > > You also mentioned that our earlier response contained a qualitative statement. To address this, we emphasize that Proposition 2.2 in our earlier response explicitly references Talagrand’s work (as cited in the provided references) and specifically analyzes the relationship between $\sup\_t X\_t$and $\mathbb{E}[\sup\_t X\_t]$.
> > > > > >
> > > > > > Given that our method is firmly grounded in and mathematically extends Talagrand’s probabilistic framework, we are unsure of the specific additional aspects you are seeking in a further comparison. Nevertheless, we will ensure that the revised version of our manuscript emphasizes this connection more clearly to avoid any potential misunderstanding.
> > > > > >
> > > > > > 3. Adding More Synthetic Experiments
> > > > > >
> > > > > > We agree that incorporating a more extensive set of synthetic experiments would better illustrate the robustness and effectiveness of our technique. However, due to time constraints (we have less than a day before the end of the rebuttal period), we are unable to provide these experiments in our current response. We will include these additional experiments in future revisions to strengthen our presentation.
> > > > > >
> > > > > > We sincerely appreciate your valuable feedback. We look forward to further refining our paper based on your detailed comments.

---

### Official Review · Reviewer_gjuP · 2024-11-02

**Soundness:** 3
**Presentation:** 3
**Contribution:** 3
**Rating:** 6
**Confidence:** 5

**Summary:**

This paper derives chaining bounds for GPR prediction intervals, inspired by Talagrand’s chaining method. This approach improves flexibility and accuracy in handling model uncertainty, overcoming issues with standard scaling of posterior deviations and assumptions of well-specified models.

**Strengths:**

1. The uncertainty quantification problem in GPR framework is very important to the community.

2. The chaining-based approach seems novel to me.

3. The bound provided in this paper is very clean, with a neat proof, and practical value.

4. The numerical results are presented clearly.

**Weaknesses:**

1. It's not clear how Theorem 1 relates to GPR bound problem. I suggest the authors to include a short discussion after Theorem 1, discussing why it's important here.

2. $t_0$ remains a magic until the concrete algorithm is presented. For instance, what's $t_0$ in GPR setup? Is it a training sample, a test sample, or something else? I suggest the authors to explain it earlier, say, after Theorem 1 or Theorem 2. Otherwise non experts may be confused when reading Section 4.

3. All examples in Figure 1 are for extrapolation (aka forecasting), but the theorem and the algorithm doesn't really depend on this setup, right? If so, can the authors either explain whether the method works for extrapolation only; or provide at least one more example to show it works for interpolation (aka infill)?

A minor issue: the font of the metrics in Section 5.2 is not consistent. For example, $PICP=$ should be $\mathrm{PICP}=$.

**Questions:**

In additional to the questions in the above weakness section, I have some additional questions.

1. Can the Theorem for RBF and Matern 3/2 be extended to a generic Matern with arbitrary smoothness? If not, can the authors briefly discuss the main challenges, and at least provide the bound for nu = 1/2 and 5/2, which are also popular in practice?

2. What happens if the domain dimension is higher? Does it impact the bound? Theoretically it seems that the answer is no, but in practical I think it matters, given the fact that GPR doesn't work really well when the domain dimension is very high, say, 100.

---

> ### Author Response · Authors · 2024-11-21
>
> Thank you very much for your comments. We address them below.
>
> Weaknesses:
>
> 1. Response:
>
> Thank you for your comment. We acknowledge that our initial writing may not have clearly conveyed the relevance of Theorem 1 to the GPR bound problem. In the revised paper, we have introduced a general bound (lines 205–215) applicable to all kernels immediately after Theorem 1. For specific kernels, such as RBF and Matérn, tighter estimates of $\mathbb{E}\left[ \sup\_{t \in T} |X\_t - X\_{t\_0}| \right]$ are derived, leading to tighter bounds, which we present along with their corresponding proofs in the subsequent subsections.
>
> Theorem 2's general bound is proven using Theorem 1, highlighting its importance. The detailed proof is provided in the appendix (lines 985-1054). Due to space constraints, we could only include this explanation in the appendix but not the main paper. However, we would try adding it in the camera-ready version if extra space is provided.
>
>
> 2. Response:
>
> Thank you for your comment. In the original manuscript, $t\_0$ was introduced in Section 4.4. In the revised paper, we have clarified its role earlier, directly following Theorem 2 (lines 204–210). Specifically, $t\_0$ is introduced to establish the bound
> $$
> \mathbb{E} \sup_{t \in T} X\_t \leq X\_{t\_0} + \mathbb{E}\left[ \sup_{t \in T} |X\_t - X\_{t\_0}| \right].
> $$
> Due to the zero-mean property of the GP and the symmetry of the covariance function, $t\_0$ is chosen such that \(X\_{t\_0}\) is close to zero; otherwise, $\mathbb{E}\left[ \sup\_{t \in T} |X\_t - X\_{t\_0}| \right]$  may be overestimated. A more detailed explanation of this choice and its implications is provided in the revised manuscript, Section 4.4 (lines 386–392).
>
> 3. Response:
>
> Thank you for your comment. Theoretically, our method relies solely on the kernel function to compute distances, enabling it to handle both extrapolation (forecasting) and interpolation (infill) tasks. This is because the kernel quantifies similarities between points based on their relative positions, making the method independent of whether the test points lie within or outside the observed range.
>
> We have clarified this in the revised paper (lines 861-891) and included an interpolation experiment using NOAA data to demonstrate this capability. In this experiment, the middle 70\% of the data was used as the test set (interpolation points), while the leftmost and rightmost 15\% were used as the training set. The results show that the predicted bounds successfully cover all test points, achieving a PICP of 1.0 and a CWC of 1.65, further supporting the effectiveness of our method for interpolation scenarios. For more details, please refer to Appendix B.5 (lines 861-891) of the revised paper.
>
> A minor issue. Response:
>
> Thank you for pointing this out. The font issue has been corrected, and the updates can be found in the revised manuscript at lines 444-447.
>
>
> Questions:
>
> 1. Response:
>
> Thank you for the question. For the general Matérn kernel with arbitrary smoothness, our proposed method provides an applicable bound. As mentioned in our response to W1, we have added a general bound after Theorem 1 (lines 204–210 of Theorem 2) in the revised manuscript, which is applicable to any kernel, including Matérn kernels with $\nu = 1/2$ and $\nu = 5/2$ that  you mentioned.
> The bounds derived for RBF and Matérn $\nu = 3/2$ are tighter than the general bound in Theorem 1 and are specifically tailored for these kernels. For other cases, such as the popular $\nu = 1/2$ and $\nu = 5/2$ of Matérn kernel, the general bound can be applied.
>
> 2. Response:
>
> Thank you for the question. Theoretically, the domain dimensionality does not directly affect the bound in our method, as GPR relies on kernel functions to compute distances. This is clarified in the revised paper (lines 254–256), where the formula for $d(s, t)$ is given:
> $$
> d(s, t)^2 = \mathbb{E}[(X\_s - X\_t)^2] = \mathbb{E}[X\_s^2] + \mathbb{E}[X\_t^2] - 2\mathbb{E}[X\_s X\_t] = K(s, s) + K(t, t) - 2K(s, t).
> $$
> Regardless of the dimensionality of the domain, the kernel function reduces these computations to a single scalar value, ensuring that the bound remains unaffected by the input dimension.

---

> > ### Comment · Reviewer_gjuP · 2024-12-01
> >
> > Thank the authors for the detailed response, and for addressing my concerns. I'll keep my positive score.

---

### Official Review · Reviewer_DSdX · 2024-11-07

**Soundness:** 2
**Presentation:** 2
**Contribution:** 3
**Rating:** 6
**Confidence:** 3

**Summary:**

The paper focuses on the problem of enhancing robustness to model misspecification in Gaussian process regression (GPR). In particular, taking important inspiration from the chaining methods of Talagrand (2014), the manuscript introduces new chaining bounds (lower and upper) for evaluating the prediction intervals of GPR. The work first revisits a decent amount of previous work, refreshes a key part of Talagrand (2014)'s contributions, derives the bounds for both RBF and Matern kernels, and finally evaluates them on a few datasets also comparing with another 3 methods.

**Strengths:**

The manuscript starts very well, with a clear description of the goals and contributions to be introduced. Later on, the review of SOTA methods and references to chaining methods in the Related work is quite enlightening. I really enjoyed reading it.

Definitely, the idea of taking chaining methods and in particular Talagrand's (2014) bound of Theorem (1) for later combining it with the Expectation of a non-negative variable Y in Eq. (3) is interesting. It is however difficult for me to evaluate the degree of the contribution here, even the novelty, but that part is a strength of the paper.

I do not particularly think that sections 4.1 and 4.2 + 4.3 are strengths of the manuscript for several reasons that I will add later in the weaknesses part. Additionally, the aspect of comparing the bounds with three recent methods is a good point, and performance seems to be on-pair or better than them (not sure if I would say "stellar" as written in L481).

**Weaknesses:**

Three thoughts that make me concerned and that I do think are points of weakness in the manuscript.

**[W1]** - The paper spends quite a lot of effort in motivating and adding context to the problem, so far the related work part is really good to me. However, after the introduction of Talagrand's method, the re-definition of RBF and Matern kernels is extremely trivial, as they are super well-known. The main problem here is that it stops the flow of explanations, and the thread of the context is lost. Later on, bounds are not really introduced but just written with two unnecessary proofs (they should really go into the Appendix). With this, I'm basically saying that bounds are not put into context and clarity is far from what is expected for a submission of this type.

**[W2]** - I'm not entirely convinced by the empirical results. Defining the performance as stellar is perhaps a bit far from humble, in my opinion, but more important than that is the analysis and evaluation of what bounds are indicating. Why is that wave-style behavior happening in Figure 1c? Is it due to the pre-training of the GP hyperparameters of some weird length-scale? I'm curious in the sense that it is different from the rest of SOTA methods.

**[W3]** - It seems that this should not be mentioned, but I have to do it. In the paper, all the scalability issues of GPR are completely ignored. In that regard, I cannot stop asking myself what is the maximum number of observations that one can consider for using Algorithm 1. Moreover, I am curious about the computational cost of such bounds, which is neither considered.

**Questions:**

Some additional questions that I thought about while reading the work.. and one minor typo:

**[Q1]** - L84: *"When many variables Xt in T are nearly identical, strong correlations between them can obscure the true variation in the process. Grouping similar variables together helps reduce this redundancy by allowing us to approximate these highly correlated variables with a representative value"* — Does this have a theoretical justification?

**[Q2]** - L97: for sufficiently large n, pi_n(t) equals t and the approximation stops.

**[Q3]** - For completeness, it would be nice to make a mention of some probabilistic bounds for GPR, i.e. Adaptive Cholesky Gaussian Processes (Bartels et al. AISTATS 2023 ) — as there is a similar spirit in some degree.

**[Minor1]** - L133 More “ecently”

---

> ### Author Response · Authors · 2024-11-21
>
> Thank you very much for your comments. We address them below.
>
> Weaknesses:
>
> [W1]. Response:
>
> Thank you for your positive feedback on the background and related work in our manuscript. We recognize that our earlier writing may have lacked coherence, potentially hindering readability and making it challenging for readers to follow the flow of our explanations.
>
> In the revised paper (uploaded), we have focused on improving logical continuity. Specifically, after introducing Talagrand’s Theorem 1, we now present a general bound earlier (lines 200-215 in the revised manuscript). This ensures that the bounds are clearly introduced early in the discussion, improving readability. The general bound applies to all GPR kernels. However, when restricted to specific kernels such as RBF and Matérn, tighter estimates of $\mathbb{E}\left[ \sup\_{t \in T} |X\_t - X\_{t\_0}| \right]$ can be derived (as discussed in Sections 4.2 and 4.3), leading to tighter bounds for these cases.
>
> Regarding the RBF and Matérn kernels, we agree that they are indeed well-known. However, we included a brief reintroduction to ensure clarity for readers who may not recall their details. Following this, we derived specific bounds for these kernels, which are tighter compared to the general bound, along with the corresponding proofs.
>
> [W2]. Response:
>
> We should have paid more attention to the tone of our writing. We will replace "stellar" with a more neutral "strong".
>
> The  wave-style behavior you mentioned in Figure 1c stems from the inherent fluctuations within the Sarcos dataset. As illustrated in the figure, all four methods exhibit similar fluctuations patterns, aligning closely with the black points that represent the actual test data. This indicates that our method is consistent with other SOTA methods in capturing the fluctuation of the dataset, rather than being different.
>
> [W3]. Response:
>
> Thank you for your insightful question. We will address this in two parts.
>
> First, regarding the limitations of GPR, the computational cost of $O(N^3)$ is a fundamental constraint inherent in GPR training,  and is not specific to our bounding method. This complexity stems from the necessity of decomposing and inverting the covariance matrix during training. It is important to emphasize that this limitation is intrinsic to the training of GPR models and thus is borne by all bounding methods that work with GPR, including $both$ the baselines and our approach.
>
> Second, we consider the computational complexity of solving the bounds in our method, and the corresponding steps are outlined in the pseudocode in the revised paper (lines 400–420). Constructing $ \{T\_n\} $ requires $ \mathcal{O}(|D\_{\text{train}}|^2 \cdot \log \log |D\_{\text{train}}|) $ time, which is dominated by kernel distance computations. Computing bounds for $ |D\_{\text{test}}| $ test points has a time complexity of $ \mathcal{O}(|D\_{\text{test}}| \cdot |D\_{\text{train}}| \cdot \log \log |D\_{\text{train}}|) $. The total computational complexity depends on the relative sizes of the training and test sets. Since the sizes of the training and test sets can vary, the overall complexity is determined by the more computationally intensive step. Thus, the total time complexity is:
> $ \mathcal{O}(\max(|D\_{\text{train}}|^2 \cdot \log \log |D\_{\text{train}}|, |D\_{\text{test}}| \cdot |D\_{\text{train}}| \cdot \log \log |D\_{\text{train}}|))$.
>
> Please note that the papers of the baseline methods do not report their computational complexity. Thus, for the purpose of comparison, we conducted runtime experiments across datasets of varying sizes, with detailed results presented in the appendix of revised paper (lines 627–671).
>
> |                  | Synthetic Data | Boston House Price | NOAA Weather | Sarcos |
> |------------------|----------------|---------------------|--------------|--------|
> | **Train Data Size** | 50             | 250                 | 255          | 250    |
> | **Test Data Size**  | 50             | 254                 | 110          | 4000   |
>
> | **Time(s)**      | Synthetic Data | Boston House Price | NOAA Weather | Sarcos |
> |------------------|----------------|---------------------|--------------|--------|
> | **RBF (Ours)**   | 0.05           | **0.52**           | 1.95         | 1.74   |
> | **Matérn (Ours)**| **0.04**       | 0.77               | **0.40**     | 1.75   |
> | **Capone22**     | 30.68          | 149.63             | 5.83         | 343.54 |
> | **Fiedler21**    | 0.07           | 0.75               | 1.34         | **1.18** |
> | **Lederer19**    | 0.56           | 2.31               | 1.92         | 2.86   |
>
> (Continued in the next comment.)

---

> > ### Author Response · Authors · 2024-11-21
> >
> > (Continuation from the previous comment.)
> >
> > Our methods (RBF and Matérn) demonstrate excellent scalability, with runtime increases that remain modest as data size grows. For example, on the large Sarcos dataset, our methods achieve runtimes of 1.74 seconds (RBF) and 1.75 seconds (Matérn), compared to Capone22 (149.63 seconds) and Lederer19 (2.31 seconds). On the smaller Boston Housing dataset, our methods achieve 0.52 seconds (RBF) and 0.77 seconds (Matérn), significantly outperforming both Capone22 and Lederer19. While Fiedler21 achieves similar efficiency on larger datasets (e.g., 1.18 seconds on Sarcos), our methods provide comparable speed while offering greater flexibility due to kernel design.
> >
> > These results support the scalability and computational efficiency of our methods, relative to the baselines, making them particularly well-suited for practical applications requiring both speed and robustness.
> >
> > Questions:
> >
> > [Q1]. Response:
> >
> > Yes, there is a clear theoretical basis for this. In deriving the bounds from Theorem 1 to Theorem 2 (line 204 of the revised manuscript), we use the formula for the expectation of non-negative random variables:
> > $ \mathbb{E} \left[ Y \right] = \int\_0^\infty P(Y \geq u) \, du $.
> > This naturally leads us to analyze $ P\left(\sup\_{t \in T} (X\_t - X\_{t\_0}) \geq u\right) $. To bound this probability, we employ the union bound:
> > $
> > P\left(\sup\_{t \in T} (X\_t - X\_{t\_0}) \geq u\right) \leq \sum\_{t \in T} P(X\_t - X\_{t\_0} \geq u).
> > $
> >
> > When many variables $(X\_t)\_{t \in T}$ are nearly identical, their corresponding events $P(X\_t - X\_{t\_0} \geq u)$ overlap significantly. This means that these variables do not contribute much independent information and instead “repeat” patterns that already exist. As a result, $\sum\_{t \in T} P(X\_t - X\_{t\_0} \geq u)$ is overestimated, making the bound loose and failing to capture the true variation within the process. Additionally, the structure of $T$ is underutilized, as the redundancy obscures the distinct contributions of individual variables.
> >
> > To address this, as highlighted on lines 85-87 of the revised paper, grouping similar variables reduces redundancy by approximating these highly correlated variables with representative values. This simplification preserves the core characteristics of the process while making the analysis more interpretable and efficient. This rationale underpins the introduction of chaining, which effectively captures the dependencies within the process and improves the tightness of the bounds.
> >
> > [Q2]. Response:
> >
> > We apologize for not understanding the question behind the reviewer’s statement. We guess that the reviewer might find the statement unclear or require further explanation.
> >
> > This statement is directly related to the chaining method's construction, as detailed in the revised paper on lines 83–107. The result that $\pi\_n(t) = t$ for sufficiently large $n$ is a consequence of this construction. Specifically, as described on lines 83–85, the decomposition $X\_t - X\_{t\_0} = \sum\_{n \geq 1} (X\_{\pi\_n(t)} - X\_{\pi\_{n-1}(t)})$ relies on this property.
> >
> > At each level $n$, the subset $T\_n \subseteq T$ refines the approximation of the original set $T$, with its diameter $\Delta(T\_n) = \sup\_{t\_1, t\_2 \in T\_n} d(t\_1, t\_2)$ shrinking as $n$ increases. This refinement ensures that for any $t \in T$, the mapping $\pi\_n(t) \in T\_n$ becomes increasingly precise as $T\_n$ becomes finer. Mathematically, as shown in lines 103–107, the iterative refinement guarantees that $d(t, \pi\_n(t)) \to 0$ as $n \to \infty$, and for sufficiently large $n$, $\pi\_n(t)$ exactly equals $t$.
> >
> > This property allows the decomposition on lines 83–85 to terminate after a finite number of steps, as the incremental terms become zero once $\pi\_n(t) = t$. Thus, the chaining method achieves both convergence and finiteness by leveraging the shrinking diameter of $T\_n$ to ensure the exact representation of $t$ at higher levels.

---

> > > ### Author Response · Authors · 2024-11-21
> > >
> > > (Continuation from the previous comment.)
> > >
> > > [Q3]. Response:
> > >
> > > Thank you for the detailed comments. Following your suggestion, we have added the reference mentioned above in the revised paper under related work on line 117.
> > >
> > > In Bartels' work, the key innovation lies in leveraging the intermediate steps of the Cholesky decomposition to bound the (expected) marginal log-likelihood of the full dataset using only a subset, instead of the entire large dataset. This approach has the practical benefit that the kernel matrix $ K $ does not need to be precomputed prior to performing the decomposition but can instead be computed on-the-fly, with little computational overhead.  As noted in our response to [W3], the computational cost of $ \mathcal{O}(N^3) $ is an inherent limitation of Gaussian processes, stemming from the need to decompose and invert the covariance matrix during training, such as in the Cholesky decomposition.
> > >
> > > This is an interesting idea, but it focuses on reducing computational overhead, which differs from our emphasis on achieving tighter bounds. That said, the idea of leveraging intermediate steps in the Cholesky decomposition might complement our method and could potentially reduce computational costs when combined. We find this perspective intriguing and may explore it in future work.
> > >
> > > [Minor1] Response:
> > >
> > > We have corrected this typo in line 135 in the revised paper (uploaded).

---

> ### Author Response · Authors · 2024-11-24
>
> Thank you for your insightful comments and valuable feedback. We have carefully considered the questions and concerns you raised and have addressed them in our revised submission and responses. We would greatly appreciate it if you could review our responses and let us know if they sufficiently address your points. Should you have any remaining questions or additional suggestions, we would be happy to address them to further enhance the paper.

---

### Meta-Review · Area_Chair_uVbC · 2024-12-20

**Metareview:**

This paper presents a novel method for computing robust bounds on predictions in Gaussian process regression based on Talagrand's chaining method and evaluates it empirically on real-world datasets against existing baselines.

Most of the reviewers found the paper well-written and clearly structured, appreciated the novel application of chaining to GPs, and acknowledged the suitability of the selected data and baselines. One reviewer raised some concerns about clarity and the degree of contribution.

The primary concerns included unclear initial explanations, discussion and contextualisation of the bounds, the lack of justification for specific empirical behaviours, and limited discussion of scalability. Additional critiques focused on the framing of "rigorous" and "deterministic" bounds and the need for statistical significance testing. One reviewer highlighted potential misalignment with the core audience of ICLR due to the paper's theoretical focus.

While many of the reviewers' concerns were addressed during the rebuttal period, Reviewer FLpc expressed that substantial adjustments need to be made before it can be considered ready for publication, and Reviewer DSdX remains concerned about the scope for ICLR community. Based on these reservations, I recommend rejecting the paper. However, I strongly recommend revising and submitting the manuscript to an upcoming, more well-scoped conference.

**Additional Comments On Reviewer Discussion:**

The review discusses the main points raised by the reviewers, and many of them were addressed in the rebuttal and post-rebuttal discussions.

---

### Decision · Program_Chairs · 2025-01-22

Reject